# SSLAM: Enhancing Self-Supervised Models with Audio Mixtures for Polyphonic Soundscapes

**Tony Alex, Sara Ahmed, Armin Mustafa, Muhammad Awais[†], Philip JB Jackson[†]**
Surrey Institute for People-Centred AI, University of Surrey, Guildford, GU2 7XH, UK
Centre for Vision, Speech and Signal Processing (CVSSP), University of Surrey
{t.alex,sara.atito,armin.mustafa,muhammad.awais,p.jackson}@surrey.ac.uk
[†]Equal senior contributions.

## Abstract

Self-supervised pre-trained audio networks have seen widespread adoption in real-world systems, particularly in multi-modal large language models. These networks are often employed in a frozen state, under the assumption that the self-supervised pre-training has sufficiently equipped them to handle real-world audio. However, a critical question remains: how well do these models actually perform in real-world conditions, where audio is typically polyphonic and complex, involving multiple overlapping sound sources? Current audio self-supervised learning (SSL) methods are often benchmarked on datasets predominantly featuring monophonic audio, such as environmental sounds, and speech. As a result, the ability of SSL models to generalize to polyphonic audio, a common characteristic in natural scenarios, remains underexplored. This limitation raises concerns about the practical robustness of SSL models in more realistic audio settings. To address this gap, we introduce Self-Supervised Learning from Audio Mixtures (SSLAM), a novel direction in audio SSL research, designed to improve the model's ability to learn from polyphonic data while maintaining strong performance on monophonic data. We thoroughly evaluate SSLAM on standard audio SSL benchmark datasets which are predominantly monophonic and conduct a comprehensive comparative analysis against state-of-the-art (SOTA) methods using a range of high-quality, publicly available polyphonic datasets. SSLAM not only improves model performance on polyphonic audio, but also maintains or exceeds performance on standard audio SSL benchmarks. Notably, it achieves up to a 3.9% improvement on the AudioSet-2M(AS-2M), reaching a mean average precision (mAP) of 50.2. For polyphonic datasets, SSLAM sets new SOTA in both linear evaluation and fine-tuning regimes with performance improvements of up to 9.1%(mAP). These results demonstrate SSLAM's effectiveness in both polyphonic and monophonic soundscapes, significantly enhancing the performance of audio SSL models. Code and pre-trained models are available at https://github.com/ta012/SSLAM.

## 1 Introduction

Self-supervised learning (SSL) has significantly advanced various domains by leveraging large volumes of unlabeled data to pre-train models effectively. In the audio domain, SSL has enabled models to achieve state-of-the-art (SOTA) performance (Gong et al., 2022; Chong et al., 2023; Huang et al., 2022a; Chen et al., 2022; Ahmed et al., 2024; Chen et al., 2022), particularly with the integration of transformer architectures (Dosovitskiy et al., 2020). Pre-trained audio models are extensively used in real-world applications, such as audio-visual segmentation (Liu et al., 2023; Park et al., 2024; Labb et al., 2024), audio captioning (Labb et al., 2024) etc. Recently, they have also been increasingly integrated into multi-modal models, particularly in multi-modal large language models (LLMs) (Zhang et al., 2023; Tang et al., 2023; Zhao et al., 2023; Panagopoulou et al., 2023). These models are often utilized in a frozen state, where a projection layer is added on top of the pre-trained

backbone to interface with systems like LLMs. In this setup, only the projector is trained, based on the assumption that the pre-trained audio models are capable of handling real-world polyphonic audio scenarios. However, the audio SSL methods are rarely evaluated for polyphonic scenarios.

Unlike monophonic audio, which features a single sound source, polyphonic audio involves auditory environments with overlapping sounds, such as musical ensembles, multi-speaker settings, or diverse natural soundscapes, like traffic and park noises. Although the term "polyphony" is often associated with music, we use it here in a broader sense to describe any audio scene with multiple concurrent sources. Given that much real-world audio is inherently polyphonic, it is essential for audio encoder representation learning to account for this complexity. While one might argue that pre-training with AudioSet (Gemmeke et al., 2017), a dataset containing multi-label samples should be adequate, it is crucial to recognize that many of the audio files in AudioSet are not truly polyphonic (refer to detailed analysis in Appendix B.1). Instead, these files often carry multiple labels that describe different facets of a single sound event with only a proportion of dataset being actually polyphonic. For example, a recording labeled as 'Carnatic music', 'Music', 'Musical instrument', and 'Classical music' all refer to the same audio event. This labeling approach does not fully encompass the complexity and richness of authentically polyphonic audio.

Given the intuitive nature of audio mixing, it would seem natural for it to serve as a standard pretext task in SSL for audio. However, this approach remains underexplored for self-supervised pre-training. Beyond its potential as a pretext task, incorporating audio mixtures into SSL offers several key advantages. First, it establishes a flexible framework for learning invariant representations that are robust to noise and generalizable across a wide range of tasks, whether involving monophonic or polyphonic audio data. Additionally, mixing audio allows for the creation of synthetic data from existing datasets, significantly increasing the diversity of training examples. This enhanced variety fosters richer learning signals, enabling the development of stronger and more semantically meaningful representations. Despite these strengths, audio mixing in SSL is still an underutilized approach, leaving substantial room for further exploration.

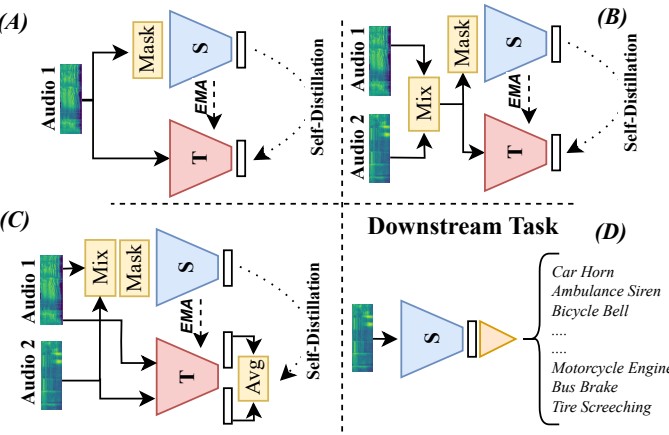

Figure 1: Overview of the components in our proposed audio SSL pre-training on unlabeled data and the audio event tagging downstream task (T: Teacher encoder, S: Student encoder). (A): Masked Latent Bootstrapping (self-distillation where the teacher is the exponentially moving average of the student) with unmixed audio (baseline). (B): Masked Latent Bootstrapping with mixed audio. (C): Source retention loss to preserve the distinct characteristics of individual audio sources. (D): Overview of the audio event tagging downstream task.

To address this limitation, we introduce SSLAM, a novel self-supervised pre-training strategy designed to enhance the ability of transformer-based models to learn from polyphonic data. In our approach, the student model receives mixed audio inputs, where multiple audio signals are randomly combined to form a polyphonic signal. Since this is a self-supervised learning method, we do not know in advance whether the selected audios are monophonic or polyphonic, meaning that the resulting audio can be either a mixture or even a mixture of mixtures. Concurrently, the teacher

model processes the same audio sources separately, averaging their features at the output of the teacher model. The student model's output is then compared to the teacher's aggregated features, enabling a robust learning mechanism that better adapts to polyphonic scenarios (refer to Figure 1).

We evaluate the proposed SSLAM on standard audio SSL benchmark datasets, including event tagging tasks covering a range of sounds, e.g. environmental sound and speech, achieving state-of-the-art performance across all categories. Furthermore, we integrate publicly available polyphonic datasets into the evaluation pipeline to assess the model's robustness in handling complex, real-world audio scenarios. We demonstrate substantial improvements in handling polyphonic audio, with our method consistently outperforming prior approaches in real-world polyphonic audio scenarios.

In summary, our contributions include:

**1.** The introduction of audio mixtures for the self-supervised pre-training, enabling the network to better adapt to real-world polyphonic audio environments.

**2.** A novel source retention loss, which explicitly preserves the individual characteristics of each audio source within the mixture. By encouraging the network to recognize and retain the distinct features of each input, this loss function ensures the integrity of each source, even when multiple sources are combined.

**3.** A comprehensive evaluation of our model on major audio SSL benchmark datasets, demonstrating SOTA performance across general audio and speech tasks compared to prior approaches.

**4.** Extensive evaluation of the model's polyphonic capabilities using several polyphonic datasets, demonstrating substantial improvements in handling real-world polyphonic audio compared to existing methods.

## 2   RELATED WORK

**Masked latent bootstrapping.** Several works  (Baade et al., 2022; Chong et al., 2023; Niizumi et al., 2022; Baevski et al., 2022; Gong et al., 2022; Ahmed et al., 2024) have explored various strategies for **masking** specific regions of the spectrogram, with pre-training primarily focused on reconstructing the masked regions within the spectrogram space. Although this approach has shown promise, other studies have questioned whether spectrogram reconstruction is the most effective method. For instance,  Chen et al. (2022) introduced the prediction of patch-level discrete labels generated by acoustic tokenizers instead of reconstructing the spectrogram. Another approach is predicting target **masked latent features** that capture higher-level, semantically meaningful information about the audio rather than focusing on low-level spectrogram reconstruction. **Bootstrapping** via EMA teacher-based self-distillation  (Grill et al., 2020; Niizumi et al., 2021) is an effective exponential moving average (EMA) approach for generating target representations. Works such as  Baevski et al. (2022; 2023); Fei et al. (2024); Chen et al. (2024) are based on the concept of **masked latent bootstrapping**. In particular,  Baevski et al. (2022; 2023); Chen et al. (2024) generate target representations by averaging outputs from multiple layers of the teacher model within an EMA-based self-distillation framework. This approach captures both high-level and low-level information, rather than relying solely on the final layer, leading to richer representations.

**Polyphonic data modeling.**   Abeßer et al. (2023) analyzed supervised audio networks such as PANN Kong et al. (2020) and PaSST Koutini et al. (2021) on low-degrees polyphony (2 events) created using ESC-50  (Piczak, 2015).   Salamon et al. (2017) introduced the Scaper library to create polyphonic datasets, with URBAN-SED  (Salamon et al., 2017) demonstrating its potential. The IDMT-DESED-FL  (Johnson et al., 2021) dataset was created using DESED  (Turpault et al., 2019) using Scaper.  SPASS is a relatively high-quality dataset created using tools such as RAVEN (Schröder, 2011), using monophonic source datasets like ESC-50 and UrbanSound8K  (Arnault et al., 2020). Networks trained with SPASS outperformed those trained on AudioSet, raising the question of whether addressing polyphony requires solely the creation of more high-quality data, or if model development should also be considered. However, to the best of our knowledge, none of the widely used audio SSL models incorporate specific design choices or training objectives aimed at addressing the challenges posed by polyphonic data.

## 3 METHODOLOGY

In this section, we outline the core components of our framework, starting with the baseline model in Section 3.1. Following, we describe the integration of audio mixtures into our SSL framework in Section 3.2, beginning with an explanation of the audio mixing strategies used to generate mixtures and pre-training the baseline model with these mixtures in Section 3.2.1. This is followed by the discussion of our proposed source retention loss, Section 3.2.2, which preserves individual source characteristics within the mixtures. Finally, we introduce our unified framework in Section 3.3, which efficiently integrates all components into a cohesive model, enabling robust learning in polyphonic as well as monophonic audio scenarios.

### 3.1 BASELINE MODEL: MASKED LATENT BOOTSTRAPPING

Our baseline leverages masked latent bootstrapping, an SSL paradigm that combines the concept of bootstrapping (Niizumi et al., 2021; Ahmed et al., 2024), spectrogram masking (Huang et al., 2022b; Baade et al., 2022), and the prediction of masked tokens in the feature space (Fei et al., 2024; Baevski et al., 2022; 2023). Approaches like Baevski et al. (2022; 2023); Chen et al. (2024) effectively combine these techniques, enabling the capture of rich contextual representations of audio data. We adopt this paradigm as the foundation of our method, aiming to enhance its performance in complex, polyphonic environment. Below, we outline the key steps of our baseline framework.

First, the input spectrogram $S$ undergoes patchification (Gong et al., 2021), where $S$ is divided into fixed-size, non-overlapping time-frequency patches. To introduce a self-supervised objective, we employ inverse block multi-masking (Baevski et al., 2023; Chen et al., 2024) with a non-masking block size of $(5 \times 5)$, where multiple masked versions of the spectrogram are generated by randomly dropping 80% of the patches from each version. A classification token (CLS) is then appended to each masked spectrogram, which is then passed through the student encoder. The student encoder processes these masked spectrograms and outputs an encoded representation $\hat{Z}$, with the CLS token's representation denoted as $\hat{Z}^{\text{CLS}}$. Following, random tokens are inserted in place of the masked tokens in the encoded representation and fed to a light-weight CNN-based decoder to predict the representations of the masked tokens, producing the student's patch-level outputs $\hat{Y}^{\text{patch}}$.

To guide the student model during training, we employ a momentum-based teacher encoder that processes the original, unmasked spectrogram. The teacher encoder's outputs are averaged across all layers to generate target representations $Z$ for the student encoder.

The training objective consists of two loss functions. First, a global loss ensures that the student model captures the overall structure of the audio signal as its mean square error (MSE):

$$\mathcal{L}_{\text{global}} = \frac{1}{B \times n_{MC}} \sum_{i=1}^{B} \sum_{j=1}^{n_{MC}} \left( \hat{Z}_{\{i,j\}}^{\text{CLS}} - Z_i^{\text{CLS}} \right)^2 \tag{1}$$

where $B$ is the batch size, $n_{MC}$ is the number of multi-mask copies, and $Z^{\text{CLS}}$ is the spatially pooled output of $Z$. Second, local loss facilitates fine-grained understanding, as below:

$$\mathcal{L}_{\text{local}} = \frac{1}{B \times n_{MC} \times |\mathcal{M}|} \sum_{i=1}^{B} \sum_{j=1}^{n_{MC}} \sum_{k \in \mathcal{M}} \left( \hat{Y}_{\{i,j,k\}}^{\text{Patch}} - Z_{\{i,k\}} \right)^2 \tag{2}$$

where $\mathcal{M}$ is the set of masked tokens.

### 3.2 SSLAM: SELF-SUPERVISED LEARNING FROM AUDIO MIXTURES

In this work, we introduce two key innovations to enhance self-supervised learning for audio. First, the introduction of audio mixtures in the SSL setting, enabling the network to better adapt to real-world polyphonic audio environments (Section 3.2.1). To complement this, we propose a novel source retention loss that explicitly preserves the individual characteristics of each audio source within the mixture. By encouraging the network to recognize and retain the distinct features of

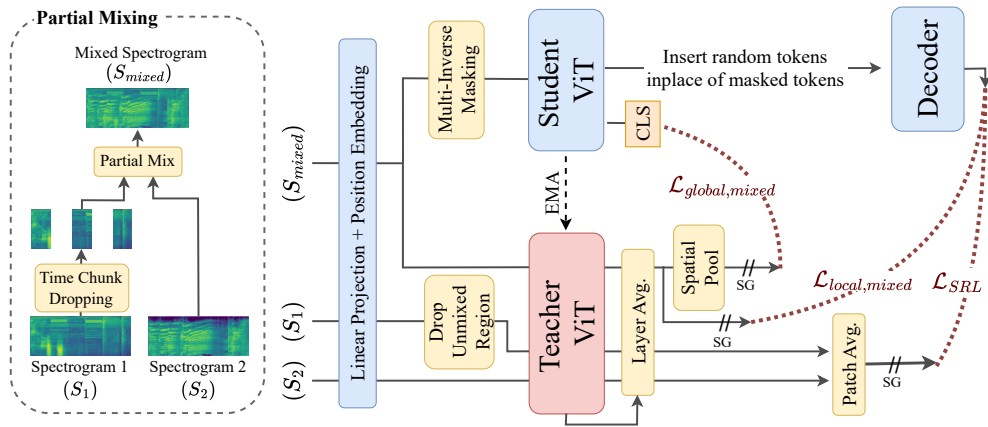

Figure 2: Illustration of novel contributions in SSLAM. The left side demonstrates the partial mixing of two audio log-mel spectrograms, while the right side visualizes the proposed novel training objectives. SG denotes the stop gradient operation applied during training.

each input, this loss function ensures the integrity of each source, even when multiple sources are combined (Section 3.2.2).

### 3.2.1 MASKED LATENT BOOTSTRAPPING USING AUDIO MIXTURES

Building on the same architecture as our baseline model, we introduce a key modification: instead of feeding the student and teacher networks a spectrogram derived from a single audio source, we provide an audio mixture spectrogram. This adjustment introduces additional complexity to the learning process.

Audio mixing can be performed either in the waveform or the spectrogram domain. Through our experiments, we found that performing the mix in the spectrogram domain yielded superior performance (refer to Appendix E.0.1). Specifically, we apply an element-wise max operation to the log-mel spectrograms of two audio signals to create an audio mixture. This technique is inspired by principles from Computational Auditory Scene Analysis (CASA), particularly the Ideal Binary Mask (IBM) (Wang, 2005), which retains the most dominant time-frequency components of overlapping signals. By using the element-wise max operation, we ensure that the most prominent time-frequency features from each source are preserved, similar to how the IBM prioritizes target-dominant regions to enhance separation and intelligibility. To formalize this, let $S_1(f, \tau)$ and $S_2(f, \tau)$ represent the log-mel-spectrograms of two audio signals, where $f$ denotes mel frequency bins and $\tau$ denotes time frames. The mixed log-mel spectrogram $S_{\text{mixed}}(f, \tau)$ is then computed using the element-wise maximum operation at each time-frequency bin, as follows:

$$S_{\text{mixed}}(f, \tau) = \max \left( S_1(f, \tau), \; S_2(f, \tau) \right), \quad \forall f, \tau. \tag{3}$$

We include sample visualizations of this approach in Appendix E.0.3. Further, additional experimental results and analysis related to this approach are provided in the Appendix E.

**Mixing vs Partial Mixing.** To achieve a balance between introducing novel audio events and preserving the main audible concepts within the original signal, we implement partial audio mixing. Instead of applying mixing across the entire audio clip, only a fraction of the audio duration is mixed. For an audio clip of length $t$, mixing is applied to 3 distinct regions, covering a total duration of $t/2$. while the original audio is preserved in the remaining $2 \times t/4$ duration (refer to Figure 2, left). By limiting the extent of mixing, we avoid overwhelming the original content, thus preserving key audio characteristics while still benefiting from the additional variability introduced by the mixing. This strategy maintains a balance between retaining essential audio features and introducing new information, ultimately leading to enhanced performance in polyphonic audio tasks.

As for the loss functions, a key modification we made when incorporating partially mixed audios into the baseline model involves the selection of teacher layers for the target representation. In the

baseline, the global loss function $\mathcal{L}_{\text{global}}$ and the local loss function $\mathcal{L}_{\text{local}}$, use the average of all 12 teacher encoder layers to form the target representation. However, with the added complexity from mixing multiple audio sources, averaging over the 12 layers then spatial pooling can potentially lead to excessive information compression, limiting the model's ability to capture meaningful global representations. To address this, we used only the final layer's output for the global loss, denoted as $\mathcal{L}_{global,mixed}$ (refer to Table 6) . For local loss $\mathcal{L}_{local,mixed}$ we used all 12 layers . It is worth to note, the optimal selection of teacher layers (top-k) may vary across modalities, as observed in Baevski et al. (2022).

### 3.2.2 SOURCE RETENTION LOSS (SRL)

To improve the model's ability to learn and retain distinct concepts from mixed audios, we introduce an additional loss function, termed Source Retention Loss (SRL). This loss encourages the model to capture the characteristics from each audio source within a mixture.

The process begins by feeding the student model with the mixed audio $S_{mixed}$, as previously described in Section 3.2.1. The student model generated representations for the mixed audio patches, which are then fed to the decoder to obtain $\hat{Y}^{\text{patch, mixed}}$.

Next, the teacher model separately processes Audio 1 spectrogram $S_1$ and Audio 2 spectrogram $S_2$. Note that, we discard the tokens from $S1$ that correspond to the unmixed regions in the partially mixed audio before passing it through the teacher model, ensuring that the teacher processes only the relevant mixed segments.

The target for the student model is created by averaging the representations produced by the teacher for $S_2$ and the processed portions of $S_1$. Finally, we employ MSE as the loss function, defined as follows:

$$
\mathcal{L}_{\text{SRL}} = \frac{1}{B \times n_{MC} \times |\mathcal{M}|} \sum_{i=1}^{B} \sum_{j=1}^{n_{MC}} \sum_{k \in \mathcal{M}} \left( \hat{Y}^{\text{patch, mixed}}_{(i,j,k)} - \left( \frac{Z^{S_2}_{(i,k)} + Z^{S_1}_{(i,k)}}{2} \right) \right)^2 \tag{4}
$$

This method encourages the network to learn multiple concepts at a granular level, thereby improving its performance in real-world polyphonic scenarios.

### 3.3 UNIFIED LEARNING FRAMEWORK

In this section, we detail how we efficiently integrate the training objectives discussed in the previous sections to construct the SSLAM framework. Our approach ensures that each objective contributes to the overall learning process without introducing unnecessary computational overhead.

To recap, the methodology consists of five objectives, The global loss for unmixed audio $\mathcal{L}_{\text{global,Unmixed}}$ captures high-level representation, while the local loss for unmixed audio $\mathcal{L}_{\text{local,Unmixed}}$ focuses on fine-grained details. For mixed audio, the global loss $\mathcal{L}_{\text{global,mixed}}$ helps the model to understand the overall structures in combined sources, and the local loss $\mathcal{L}_{\text{local,mixed}}$ understands overlapping sounds at a granular level. Finally, the source retention loss $\mathcal{L}_{\text{SRL}}$ ensures that distinct characteristics of each source are preserved in the mixed audio. For clarity, Figure 2 highlights only the novel contributions of our SSLAM framework.

To train the model, we first pre-train it using only unmixed audio using the $\mathcal{L}_{\text{global,Unmixed}}$ and $\mathcal{L}_{\text{local,Unmixed}}$ loss functions. This step allows the network to learn foundational representations of distinct audio events and establish robust feature extraction capabilities without the added complexity of mixed signals. We refer to this as "Stage 1".

After this, in "Stage 2", the model is trained on a combination of partially mixed audio (half the batch) and unmixed audio (the other half), utilizing all five losses. This two-stage training approach allows the model to progressively develop the capacity to handle more complex polyphonic audio. The inclusion of all five training objectives in "Stage 2" is motivated as follows: Using unmixed audio based training objectives enhances the robustness of our approach on monophonic datasets (refer to Appendix D) while enabling the network to learn foundational representations enabling robust feature extraction capability without the added complexity of mixed signals. Incorporat-

ing mixed audio based training objectives exposes the model to diverse polyphonic data, thereby improving its performance on polyphonic tasks. Additionally, the source retention loss (SRL), effectively computed by leveraging the two mixed and unmixed halves of the batch, explicitly ensures that the representation of mixed audio stays true to its source components. This further enhances the network's ability to understand polyphonic audio.

Finally, these objectives are efficiently incorporated into the SSLAM framework as follows:

---

**Algorithm 1** Efficient Incorporation of Training Objectives

---

1: **Input:** A batch of log-mel spectrograms $B$
2: **Step 1:** Create a partially mixed batch $B_m$ by rolling and mixing $B$ along the batch dimension.
3: **Step 2:** Concatenate $B$ and $B_m$ to form a combined batch $2B$.
4: **Step 3:** Forward $2B$ through the student and teacher networks, reducing the number of multitask clones from 16 to 8 for consistency with the baseline.
5: **Step 4:** For SRL, mask and drop unmixed regions in $B$ post-positional embedding and forward the result to the teacher.
6: **Step 5:** Compute the five training objectives using the relevant parts of the batches.

---

This process enables the seamless integration of both unmixed and partially mixed audio, along with their respective training objectives, ensuring SSLAM's adaptability and effectiveness across both monophonic and polyphonic audio datasets.

## 4 EXPERIMENTS

### 4.1 DATASETS

For pre-training, we utilized the AS-2M dataset without any label information. For downstream evaluation, we employed various audio SSL benchmark datasets, including AS-2M, AS-20K, ESC-50, KS1, and KS2, as well as polyphonic datasets such as SPASS, IDMT-DESED-FL, and URBAN-SED. More information about these datasets can be found in Appendix B.

### 4.2 IMPLEMENTATION DETAILS

**Patchification and Positional Encoding.** In the SSLAM framework, input spectrograms are divided into non-overlapping patches using a CNN layer with a kernel size of (16,16) and a stride of 16, followed by the addition of positional encoding to retain information about the order of the patches.

**Encoder.** We used ViT-Base model (Dosovitskiy et al., 2020) for both the student and teacher encoders. The teacher model is an Exponential Moving Average (EMA) version of the student model. The teacher's parameters, $\theta_t$, are updated according to the EMA rule:

$$\theta_t \leftarrow \tau\theta_t + (1 - \tau)\theta_s \tag{5}$$

where $\theta_s$ represents the student model's parameters. Initially, the momentum term $\tau$ is set low to allow greater flexibility during early training. As training progresses, $\tau$ gradually increases toward 1, ensuring more stable updates and convergence between the student and teacher models.

**Decoder.** To decode the masked patches, we employ a lightweight 6-layer network comprising 2D CNN layers, LayerNorm, and GELU activation.

The total number of parameters used was 93M during pre-training and 88M during fine-tuning.

### 4.3 PRE-TRAINING DETAILS

**Pre-Training.** For pre-training we used only the AS-2M dataset. Input waveforms were uniformly resampled to 16kHz and transformed into 128-dimensional mel-frequency bands using a 25ms Hanning window and a 10ms hop size.

**Stage 1**: The model was initialized from scratch and trained for 10 epochs using training objectives $\mathcal{L}_{global,Unmixed}$ and $\mathcal{L}_{local,Unmixed}$ as described in Section 3.3.

**Stage 2**: The model was initialized with the pre-trained weights from Stage 1 and further trained for 5 epochs using all the training objectives outlined in Section 3.3.

All pre-training experiments were conducted on $4\times$ Nvidia 3090 GPUs, with each epoch taking 7 hours in Stage 1 and 7.5 hours in Stage 2.

**Component-wise analysis of SSLAM framework.** To better understand the contribution of each component in helping the model in understanding polyphonic audio, we developed four variants of our approach, incrementally building the SSLAM framework during Stage 2 of pre-training. These variants are as follows: 1.Masked latent bootstrapping using unmixed audio (MB-UA): uses only $\mathcal{L}_{global,Unmixed}$ and $\mathcal{L}_{local,Unmixed}$. 2.Masked latent bootstrapping using partially mixed audio (MB-PMA): uses $\mathcal{L}_{global,mixed}$ and $\mathcal{L}_{local,mixed}$. 3.Masked latent bootstrapping using unmixed and partially mixed audio (MB-UA-PMA): uses $\mathcal{L}_{global,Unmixed}$, $\mathcal{L}_{local,Unmixed}$, $\mathcal{L}_{global,mixed}$ and $\mathcal{L}_{local,mixed}$. 4. Source Retention Loss + Masked latent bootstrapping using unmixed and partially mixed audio (SSLAM): This final variant integrates the source retention loss with the other training objectives.

For a fair comparison, we pre-trained all four models on the AS-2M dataset with a batch size of 48 for 5 epochs as part of Stage 2 pre-training (refer to the AppendixA for further details). We evaluate how performance varies across various polyphonic soundscape datasets (refer to Table 2), including AS-20K, and across different degrees of polyphony (refer to Table 3).

**Downstream Task Training.** We conducted a comparative analysis of our approach on standard audio SSL benchmark datasets, alongside prior methods (refer to Table 1). To further assess polyphony handling, we evaluated the model on different polyphonic datasets as discussed before (refer to Table 2.) All downstream tasks, except for AS-2M, were trained using $1\times$ Nvidia 3090 GPU, while AS-2M used $1\times$ Nvidia A100 GPU (refer to Appendix A for further details).

## 4.4 EVALUATION CRITERION

**Fine-tuning.** We assess the effectiveness of our proposed approach by fine-tuning, where the entire network is trained on downstream tasks with labeled data.

**Linear evaluation.** We use linear evaluation (also referred to as linear probing) to better assess the quality of the representations learned through self-supervised learning. In this method, a linear classifier is trained on top of the frozen pre-trained representations, preventing the rest of the network from updating. This provides a clearer measure of the intrinsic quality of the learned features, without task-specific adaptation, making it a more reliable evaluation of SSL performance than fine-tuning. Moreover, since pre-trained networks are often used in a frozen state in real-world applications, linear evaluation offers a more practical and insightful assessment of the model's general utility across diverse tasks.

## 5 PERFORMANCE DISCUSSION

**Audio SSL benchmark datasets.** Our approach demonstrated significant performance improvements across most audio SSL benchmark datasets (refer to Table 1), with a notable gain of 3.9% relative improvement over the previous state-of-the-art on AS-2M, reaching a mAP of 50.2. Among the purely monophonic datasets, such as ESC-50, KS2, and KS1, our model demonstrated comparable performance to the SOTA, which varies across different datasets. This underscores the universality and robustness of our approach, as it performs consistently across a variety of monophonic and general audio datasets.

**Polyphonic audio datasets** Overall, both Table 2 and Table 3 demonstrate that incorporating partially mixed data (MB-PMA) consistently improves performance compared to the baseline MB-UA. These results highlight that our mixing strategy, by exposing the model to mixed audio, increases its access to diverse polyphonic data that AudioSet alone cannot sufficiently provide. The slight performance decrease seen in MB-UA-PMA is expected, as only half the batch is used for partially mixed data to accommodate the unmixed data. However, this design is crucial for handling both types of data within the framework and for the efficient integration of the SRL, which ultimately enables

Table 1: Evaluation on audio SSL benchmark datasets compared to previous methods. The pre-training datasets include AudioSet (AS), and LibriSpeech (LS). To enhance clarity, methods utilizing additional supervised training on external datasets are *grayed out*. For AS-2M and AS-20K, mean average precision (mAP) is used as the evaluation metric, while classification accuracy is reported for all other datasets. For more details about the datasets refer to Appendix B.0.1.

| Model | #Param | Pre-training Data | Audio | | | Speech | |
| --- | --- | --- | --- | --- | --- | --- | --- |
| | | | AS-2M | AS-20K | ESC-50 | KS2 | KS1 |
| SS-AST (Gong et al., 2022) | 89M | AS+LS | - | 31.0 | 88.8 | 98.0 | 96.0 |
| MAE-AST (Baade et al., 2022) | 86M | AS+LS | - | 30.6 | 90.0 | 97.9 | 95.8 |
| MaskSpec (Chong et al., 2023) | 86M | AS | 47.1 | 32.3 | 89.6 | 97.7 | - |
| MSM-MAE (Niizumi et al., 2022) | 86M | AS | - | - | 85.6 | 87.3 | - |
| data2vec (Baevski et al., 2022) | 94M | AS | - | 34.5 | - | - | - |
| Audio-MAE (Huang et al., 2022a) | 86M | AS | 47.3 | 37.1 | 94.1 | 98.3 | 96.9 |
| BEATs$_{iter3}$ (Chen et al., 2022) | 90M | AS | 48.0 | 38.3 | 95.6 | 98.3 | 97.7 |
| BEATs$_{iter3+}$ | 90M | AS | 48.6 | 38.9 | 98.1 | 98.1 | 98.1 |
| ASiT (Ahmed et al., 2024) | 86M | AS | 48.0 | 38.6 | 95.3 | **98.9** | 98.2 |
| A-JEPA (Fei et al., 2024) | 86M | AS | 48.6 | 38.4 | **96.3** | 98.5 | 97.7 |
| EAT (Chen et al., 2024) | 88M | AS | 48.6 | 40.2 | 95.9 | 98.3 | - |
| **SSLAM (Ours)** | 88M | AS | **50.2** | **40.9** | 96.2 | 98.1 | **98.8** |

Table 2: Impact of individual novel contributions evaluated across various polyphonic datasets. All performances are reported in mAP. For more details about the datasets refer to Appendix B.0.2.

| Model | SPASS | | | | | IDMT DESED | URBAN SED | AS-20K |
| --- | --- | --- | --- | --- | --- | --- | --- | --- |
| | Square | Park | Waterfront | Street | Market | | | |
| **Linear Evaluation** | | | | | | | | |
| MB-UA | 60.1 | 59.7 | 55.2 | 63.7 | 62.8 | 75.8 | 71.3 | 13.9 |
| MB-PMA (Ours) | 63.1 | 63.5 | 58.5 | 66.5 | 67.4 | 78.4 | 70.9 | 16.1 |
| MB-UA-PMA (Ours) | 62.7 | 63.5 | 58.2 | 66.6 | 66.6 | 77.7 | 70.9 | 15.2 |
| SSLAM (Ours) | **64.2** | **64.2** | **59.5** | **67.4** | **68.5** | **77.8** | **71.4** | **16.9** |
| **Fine-tuning** | | | | | | | | |
| MB-UA | 84.4 | 78.4 | 80.1 | 81.4 | 89.7 | 94.4 | **90.9** | 40.4 |
| MB-PMA (Ours) | 85.1 | 80.0 | 82.0 | 82.2 | 90.8 | 94.4 | **90.9** | 40.6 |
| MB-UA-PMA (Ours) | 85.0 | 79.7 | 82.0 | 82.2 | 90.5 | 94.4 | **90.9** | 40.7 |
| SSLAM (Ours) | **85.6** | **80.5** | **82.6** | **82.2** | 90.2 | **94.5** | 90.9 | **40.9** |

SSLAM to surpass all other models, achieving a performance improvement of up to 9.1% on SPASS (Market). The performance gap is particularly evident in the linear evaluation setting, suggesting that our contributions have enabled SSLAM to learn representations that generalize more effectively to polyphonic data compared to other approaches. Upon examining the tables individually, Table 2 shows that the proposed approach consistently outperforms the baseline across all datasets, achieving improvements of up to 9.1%, with a significant jump of 21.6% in AS-20K. In Table 3, although performance slightly decreased for lower polyphony level ({2,3}) in the linear evaluation setting, and the fine-tuning improvements were marginal, the performance gap widened as we moved to medium and higher polyphony levels, reaching up to 9.7% ({8,9}).

**Additional ablations.** As we have already analyzed the relevance of each of the components of the approach previous section, here we discuss additional ablations related to our approach. All the experiments discussed in this section are evaluated on downstream task AS-20K in the fine-tuning regime. The following are our observations. We observed that top k for teacher layer averaging is 1 for global loss and 12 for local loss (refer to Table 6); in regard to the extent of spectrogram mixing, partial mixing was found to be better than full mixing (refer to Table 4); as local loss contributed the foundation of the mask latent boostrapping approach, we investigated the effect of global loss with unmixed data, mixed data and SRL. Our experiment showed that everywhere except SRL, the global loss showed performance improvement (refer to Table 5).

Table 3: Evaluation on the *Degrees of polyphony* dataset: Assessing the impact of various individual contributions across different polyphony levels. $\{a,b\}$ denotes a data subset where audio files contain $a$ or $b$ distinct sound events. All performances are reported in mAP. For more details about the datasets refer to Appendix B.0.2.

| Model | Unmixed Data | Partial Mixed Data | SRL | Number of Distinct sound events | | | | | | |
| | | | | {2,3} | {4,5} | {6,7} | {8,9} | {10,11} | {12,13} | {14+} |
| --- | --- | --- | --- | --- | --- | --- | --- | --- | --- | --- |
| **Linear Evaluation** | | | | | | | | | | |
| MB-UA | ✓ | ✗ | ✗ | **61.5** | 69.4 | 45.8 | 53.5 | 58.3 | 61.6 | 66.7 |
| MB-PMA (Ours) | ✗ | ✓ | ✗ | 58.6 | 70.0 | 50.7 | 57.2 | 61.3 | 64.8 | 67.6 |
| MB-UA-PMA (Ours) | ✓ | ✓ | ✗ | 58.2 | 70.0 | 49.8 | 56.9 | 61.1 | 64.7 | 67.9 |
| SSLAM (Ours) | ✓ | ✓ | ✓ | 60.6 | **70.6** | **53.2** | **58.7** | **63.0** | **66.1** | **69.7** |
| **Fine-tuning** | | | | | | | | | | |
| MB-UA | ✓ | ✗ | ✗ | 87.3 | 86.5 | 69.5 | 81.5 | 82.5 | 80.7 | 78.1 |
| MB-PMA (Ours) | ✗ | ✓ | ✗ | 87.3 | **86.9** | 71.4 | 83.0 | 83.4 | 82.0 | 79.3 |
| MB-UA-PMA (Ours) | ✓ | ✓ | ✗ | 87.2 | 86.4 | 70.3 | 82.7 | 83.4 | 81.8 | 78.8 |
| SSLAM (Ours) | ✓ | ✓ | ✓ | **87.7** | **86.9** | **71.9** | **83.3** | **83.8** | **82.2** | **79.4** |

Table 4: Comparison of partial vs. full mixing of mel-spectrograms via element wise max operation, evaluated across Stage 1 and Stage 2 of our curriculum. All models are exclusively trained with the specified data, e.g. models with tag "mixed audio" were trained solely on batches of mixed audio.

| Model | mAP |
| --- | --- |
| **Finetuning** | |
| Stage 1 with unmixed audio | **40.2** |
| Stage 1 with mixed audio | 39.0 |
| Stage 1 with mixed audio 75% of updates | 39.4 |
| Stage 1 with partial mixed audio | 39.9 |
| Stage 2 with unmixed audio | 40.4 |
| Stage 2 with mixed audio 75% of updates | 40.4 |
| Stage 2 with partial mixed audio | **40.6** |

Table 5: Effect of global loss

| Local loss | | | Global loss | | | AS-20K |
| UM | PM | SRL | UM | PM | SRL | |
| --- | --- | --- | --- | --- | --- | --- |
| ✓ | ✓ | ✓ | ✗ | ✗ | ✗ | 40.4 |
| ✓ | ✓ | ✓ | ✓ | ✗ | ✗ | 40.7 |
| ✓ | ✓ | ✓ | ✓ | ✓ | ✗ | **40.9** |
| ✓ | ✓ | ✓ | ✓ | ✓ | ✓ | 40.6 |

Table 6: Top k layers

| Model | mAP |
| --- | --- |
| **Fine-tuning** | |
| Global:12, Local:12 | 40.5 |
| Global:1, Local:12 | **40.6** |
| Global:3, Local:3 | 39.3 |
| Global:1, Local:3 | 39.4 |
| Global:1, Local:1 | 38.8 |

# 6 CONCLUSION

Despite the rapid advancements in audio self-supervised learning (SSL) over recent years, the ability of audio SSL models to effectively handle polyphonic audio remains underexplored. This is a significant concern, as most real-world sounds are inherently polyphonic. In this work, we proposed the SSLAM framework, an audio SSL pre-training approach aimed at enhancing the model's ability to handle polyphonic audio, while maintaining strong performance on monophonic data. The framework enables the model to learn from audio mixtures employing novel training objectives, including the Source Retention Loss. Additionally, we expanded the existing audio SSL benchmarks, which have predominantly focused on monophonic datasets, by incorporating diverse polyphonic datasets. Our proposed approach demonstrated state-of-the-art performance on both traditional audio SSL benchmark datasets and the newly included polyphonic datasets.

## ACKNOWLEDGMENTS

This research was supported by the EPSRC-BBC Prosperity Partnership 'AI4ME: Future Personalised Object-Based Media Experiences Delivered at Scale Anywhere' (EP/V038087/1). We also

acknowledge the use of computational resources on the Tier 2 HPC facility JADE2, funded by EP-SRC (EP/T022205/1). Additionally, we thank Srinivasa Nandam for the valuable discussions on self-supervised learning.

## ETHICS STATEMENT

We relied on publicly available datasets for all of our experiments. While the proposed approach holds promise for a variety of beneficial applications, there is a possibility it could be misused for purposes such as surveillance. We are mindful of these risks and will ensure that the distribution of our code and models is handled with caution and ethical consideration.

## REPRODUCIBILITY STATEMENT

We document all implementation details in Section 4.2, pre-training details in 4.3 and Appendix A. Code and pre-trained models are available at `https://github.com/ta012/SSLAM`

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

## A    TRAINING HYPER-PARAMETERS

Additional hyper-parameters used in pre-training using AS-2M and fine-tuning of standard audio SSL benchmark datasets are listed in Table 7.  and that for evaluation of polyphonic datasets are listed in Table 8

Table 7: SSLAM pre-training and audio SSL benchmark dataset fine-tuning hyper-parameters.

| Hyperparameters | Pre-Training | Fine-Tuning | | | | |
|---|---|---|---|---|---|---|
| | AS-2M | AS-2M | AS-20K | ESC-50 | KS1 | KS2 |
| | Stage 1 \| Stage 2 | | | | | |
| Optimizer | AdamW (Loshchilov & Hutter, 2017) | | | | | |
| Optimizer Momentum | $\beta_1 = 0.9, \beta_2 = 0.95$ | | | | | |
| Weight Decay | 0.05 | | | | | |
| Learning Rate Schedule | Cosine (Loshchilov & Hutter, 2016) | | | | | |
| Peak Learning Rate | 0.0005\|0.00005 | 0.00005 | 0.00005 | 0.0001 | 0.0002 | 0.0002 |
| Minimum Learning Rate | 0.000001 | | | | | |
| Steps | 400K\|200K | 400K | 40K | 8K | 60K | 60K |
| Warm-up steps | 53K\|25K | 40K | 4K | 800 | 6K | 6K |
| Batch size | 12 | 96 | 48 | 48 | 256 | 256 |
| Clone batch | 16\|8 | | | N/A | | |
| Number of GPUs | 4 | | | 1 | | |
| Dropout (Srivastava et al., 2014) | 0.0 | 0.0 | 0.0 | 0.0 | 0.0 | 0.0 |
| Drop path (Huang et al., 2016) | 0.0 | 0.1 | 0.1 | 0.1 | 0.1 | 0.1 |
| Weighted Sampling | False | True | False | False | False | False |
| Weighted Sampling size | N/A | 200K | N/A | N/A | N/A | N/A |
| Roll Augmentation | False | True | True | True | False | False |
| Noise Augmentation | False | False | False | False | True | True |
| SpecAug (Park et al., 2019) | N/A | 0.2 | 0.2 | 0.2 | 0.1 | 0.1 |
| Mixup (Zhang et al., 2017) | 0.0 | 0.8 | 0.8 | 0.0 | 0.8 | 0.8 |
| Multilabel | N/A | True | True | False | False | False |
| Loss Function | MSE | BCE | BCE | CE | BCE | BCE |
| Dataset Mean for Normalization | -4.268 | -4.268 | -4.268 | -6.627 | -6.846 | -6.846 |
| Dataset Std for Normalization | 4.569 | 4.569 | 4.569 | 5.359 | 5.565 | 5.565 |

Table 8: SSLAM polyphonic datasets linear evaluation and fine-tuning hyper-parameters.

| Hyperparameters | Linear Evaluation\|Fine-Tuning | | | | |
|---|---|---|---|---|---|
| | SPASS SUBSETS | IDMT | URBAN SED | Degrees of Polyphony Dataset | AS-20K |
| Optimizer | AdamW (Loshchilov & Hutter, 2017) | | | | |
| Optimizer Momentum | $\beta_1 = 0.9, \beta_2 = 0.95$ | | | | |
| Weight Decay | 0.05 | | | | |
| Learning Rate Schedule | Cosine (Loshchilov & Hutter, 2016) | | | | |
| Peak Learning Rate | 0.001\|0.00005 | | | | |
| Minimum Learning Rate | 0.000001 | | | | |
| Epochs | 50 | 50 | 50 | 50 | 94 |
| Warm-up epochs | 5 | 5 | 5 | 5 | 10 |
| Batch size | 48 | 48 | 48 | 48 | 48 |
| GPUs | 1 | 1 | 1 | 1 | 1 |
| Dropout (Srivastava et al., 2014) | 0 | 0 | 0 | 0 | 0 |
| Drop path (Huang et al., 2016) | 0.1 | 0.1 | 0.1 | 0.1 | 0.1 |
| Roll Augmentation | True | True | True | True | True |
| SpecAug (Park et al., 2019) | 0.2 | 0.2 | 0.2 | 0.2 | 0.2 |
| Mixup (Zhang et al., 2017) | 0.8 | 0.8 | 0.8 | 0.8 | 0.8 |
| Multilabel | True | True | True | True | True |
| Loss Function | BCE | BCE | BCE | BCE | BCE |
| Dataset Mean for Normalization | -5.275 | -5.464 | -5.561 | -5.216\|-5.659 | -4.268 |
| Dataset Std for Normalization | 3.268 | 3.380 | 2.699 | 3.376\|2.620 | 4.569 |

# B DATASET DETAILS

### B.0.1 COMMONLY USED AUDIO SSL BENCHMARK DATASETS

**AudioSet** (Gemmeke et al., 2017) is a large-scale dataset containing over 2 million 10-second audio clips sourced from YouTube, annotated with 527 audio event classes. The dataset is divided into three subsets: a balanced set (∼20k clips), an unbalanced set (around 2M files), and an evaluation set (roughly 20k files). Since the dataset is dynamically sourced from YouTube, its availability may decrease over time as videos are removed or taken down. Our downloaded and processed copy of AudioSet includes 1.91M files in unbalanced, 21K in balanced, and 19K in the evaluation set. For pre-training, we use the full dataset by combining the unbalanced and balanced sets, referred to as AudioSet-2M (AS-2M) in this paper, while the balanced set alone is referred to as AS-20K. No label information is used during pre-training

**Environmental Sound Classification (ESC-50)** (Piczak, 2015) is a collection of 2000, 5-second environmental sound recordings across 50 classes. Each recording is annotated with a single class. Following previous works Chen et al. (2024; 2022); He et al. (2022), we employ a 5-fold cross-validation setting and report the classification accuracy as the evaluation metric.

**Speech Commands (KS1, KS2)** (Warden, 2018) are datasets designed for keyword spotting tasks. KS2 consists of 105,829 1-second recordings across 35 distinct speech commands, with the dataset split into 84,843 samples for training, 9,981 samples for validation, and 11,005 samples for testing. KS1, an earlier version of the dataset, contains 12 classes: 10 specific keyword classes, 1 silence class, and 1 unknown class that includes samples from 20 additional speech commands not explicitly covered by the keyword set. In line with previous works, we train models on the training split, select the best-performing model based on validation, and report test results. The evaluation metric used is classification accuracy.

### B.0.2 POLYPHONIC AUDIO DATASETS

**SPASS** is a high-quality synthetic polyphonic dataset designed for sound event detection (SED) with a particular focus on spatiotemporal labeling of sound sources. The dataset was generated using acoustic virtual reality tools such as RAVEN (Schröder, 2011) and monophonic source datasets like ESC-50 and UrbanSound8K. It simulates the following five distinct urban soundscapes square, park, waterfront, street, and market, ensuring that the dataset remains general and not confined to a specific environment. Each soundscape contains approximately 3,750 samples in the training split and 1,250 samples in the evaluation split.

**IDMT-DESED-FL** (Johnson et al., 2021) dataset was created using scraper tool with data from DESED (Turpault et al., 2019) for sound event detection. The training split contains 10,000 audio files, while the evaluation split includes 2,000 files.

**URBAN-SED** (Salamon et al., 2017) introduced the Scaper library to create polyphonic datasets, with URBAN-SED demonstrating its potential. As we are using this dataset to evaluate polyphonic capability, we only included audio files with more than one label. After applying this filter, the training set consists of 5,268 audio files, and the evaluation set contains 1,739 files.

**Degrees of Polyphony** We created dataset with varying degrees of polyphony, ranging from 2 to 14+, using audio files from SPASS and URBAN-SED, which we will be referring to as the *Degrees of Polyphony* dataset in this paper. This allows us to evaluate the model's performance across different levels of polyphony.

### B.1 IS THE AUDIOSET DATASET TRULY POLYPHONIC? AN ANALYSIS

AudioSet (Gemmeke et al., 2017) is one of the largest collections of multi-labeled audio files. While the term "multi-label" may suggest the presence of multiple distinct sound events within a single audio file, implying polyphony, this is not always the case. As discussed in the introduction, labels such as 'Carnatic music,' 'Music,' 'Musical instrument,' and 'Classical music' often refer to the same underlying sound event. Despite these labels appearing together, the audio file may feature only a single distinct sound, thereby giving a misleading impression of polyphony.

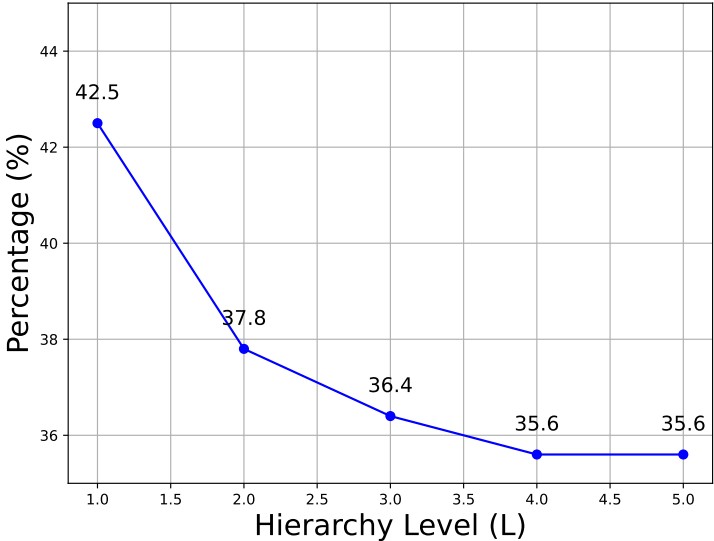

Figure 3: Percentage of audio files in AudioSet (Gemmeke et al., 2017) with at least 2 distinct sound events (y-axis) based on our evaluation criteria, at different values of hierarchy level $L$ (x-axis).

To assess the extent of this and to analyze the proportion of audio files with truly distinct sound events, we conducted an analysis using the ontology and label information provided with AudioSet.

The steps in the process are detailed below:

---

**Algorithm 2** AudioSet Identify Files with Distinct Sound Events

---

1: **Input:** Ontology file, AudioSet label data, hierarchy level $L$
2: **Step 1:** Parse the ontology file to create mappings for `id_to_parent` and `id_to_children`.
3: **Step 2:** For each audio file:
  - Retrieve all associated labels.
  - For each label, find all related labels (parents or children) up to the hierarchy level $L$.
  - Remove these related labels from the label set to retain only the distinct sound events.
4: **Output:** Percentage of audio files with at least 2 distinct sound events.

---

In our analysis, we found that even when considering only one level of the hierarchy, only 42.5% of audio files contain at least two distinct sound events. This number decreases as we explore deeper levels of the hierarchy. The motivation for considering levels greater than one ($L > 1$) is illustrated by examples such as "Wild animals" $\rightarrow$ "Roaring cats (lions, tigers)" $\rightarrow$ "Roar." In cases where labels like "Roar" and "Wild animals" appear together, they may still refer to the same sound event. Please refer to Figure 3 for the "percentage of audio files with distinct sound events" based on our evaluation criteria described above, at different values of $L$.

As the hierarchy level increases from 1 to 4 the percentage of audio files containing at least two distinct sound events (labels) progressively decreases: from 42.5% at level 1 to 37.8% at level 2, 36.4% at level 3, and finally 35.6% at level 4. This substantiates our argument that relying solely on AudioSet is insufficient for developing models capable of handling polyphonic audio effectively.

## C  REPRESENTATION LEARNING VIA CONCEPT SEPARATION WITH MIXTURE INVARIANT LOSS

In the development of SSLAM, we investigated whether separating multiple concepts from mixed audio inputs to the student model could enhance polyphonic audio understanding. To achieve this, we employed the Mixture Invariant Training (MixIT) loss (Wisdom et al., 2020), originally proposed for audio source separation tasks. Specifically, we extended the student encoder with an additional MLP block to project its output representations into $K$ distinct representation vectors.

The general overview of the process is as follows. Given a mixed audio input $S_{\text{mixed}}$ provided to the student model, its output representation is further projected by an MLP into $K$ distinct representation vectors, collectively represented as $\hat{\mathbf{Y}}_{\text{mixed}} = \{\hat{Y}_{\text{mixed}}^{(1)}, \hat{Y}_{\text{mixed}}^{(2)}, \ldots, \hat{Y}_{\text{mixed}}^{(K)}\}$. This projection aims to separate the mixed audio representation into its constituent representations or concepts. In parallel, the teacher model processes individual spectrograms $S_1$ and $S_2$ corresponding to the constituent audio signals, producing outputs $Z^{S_1}$ and $Z^{S_2}$, respectively (refer to Figure 4).

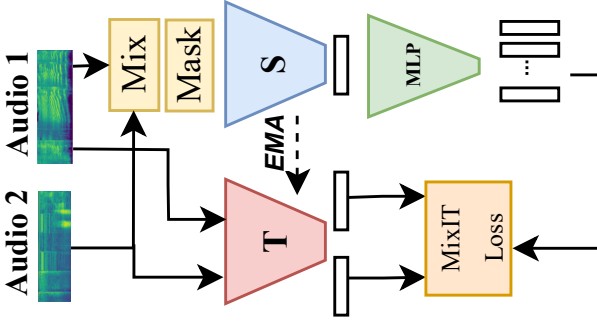

Figure 4: Overview of concept separation using mixture invariant training loss

To enforce concept separation, we use the MixIT loss, which is defined as follows:

$$\mathcal{L}_{\text{MixIT}}\left(Z^{S_1}, Z^{S_2}, \hat{\mathbf{Y}}_{\text{mixed}}\right) = \min_{\mathbf{A}} \sum_{i=1}^{2} \mathcal{L}\left(Z^{S_i}, \left[\mathbf{A}\hat{\mathbf{Y}}_{\text{mixed}}\right]_i\right), \tag{6}$$

where we employ Mean Squared Error (MSE) as $\mathcal{L}$. The *mixing matrix* $\mathbf{A} \in \mathbb{B}^{2 \times K}$ is a binary matrix constrained such that each column sums to 1. This assigns each of the $K$ representation vectors to either $Z^{S_1}$ or $Z^{S_2}$. In our initial experiments, we observed that this approach yielded worse performance compared to SSLAM on the AS-20K benchmark (39.9 mAP vs. 40.9 mAP).

One key drawback of this approach is that mixture invariant training assumes the independence of individual sources, which is a reasonable assumption in the case of audio source separation task. However, in the representation (concept) space, this assumption is less valid, as there is a high likelihood that one representation overlaps with both individual audio inputs. This overlap undermines the independence constraint. Additionally, the number of possible partitions grows exponentially with $K$, leading to $2^K$ potential loss terms. This exponential growth introduces significant computational complexity, making the approach less practical for larger $K$.

Despite these challenges, we believe that concept separation remains a promising avenue for polyphonic audio understanding, warranting further studies to address these limitations and refine the approach.

## D    REASONING BEHIND INCORPORATING UNMIXED DATA IN MB-UA-PMA VARIANT

In Tables 2 and 3, the MB-UA-PMA variant was developed by splitting the batch in MB-PMA into two halves: one for unmixed audio and the other for mixed audio, rather than using the entire batch for mixed audio. While this modification leads to a decrease in performance on polyphonic audio datasets (refer to Tables 2 and 3), it ensures that SSLAM achieves robustness across both monophonic and polyphonic audio datasets.

Empirical evidence supporting this design choice is provided in Table 9, where MB-PMA and MB-UA-PMA are evaluated on monophonic audio datasets, such as ESC-50 and KS2, as well as the AS-20K dataset. The improved performance of MB-UA-PMA with respect to MB-PMA in Table D and the decreased performance in Tables 2 and 3 highlight the trade-off between achieving higher performance on polyphonic datasets and promoting better generalization across diverse audio data.

Additionally, the performance reduction observed for MB-UA-PMA in Tables 2 and 3 is mitigated in our final variant, SSLAM, through the introduction of the Source Retention Loss (SRL), which utilizes the unmixed half of the batch for its computation (refer to Algorithm 1).

Table 9: Comaprision of MB-PMA and MB-UA-PMA on monophonic and AS-20K datasets

| Model | ESC-50 | KS2 | AS-20K |
|---|---|---|---|
| **Fine-tuning** | | | |
| MB-PMA | 95.5 | 97.9 | 40.6 |
| MB-UA-PMA | **96.2** | **98.0** | **40.7** |

## E    ADDITIONAL INFORMATION ON AUDIO MIXING

### E.0.1    DIFFERENT INPUT AUDIO MIXING STRATEGIES

We explored several audio mixing strategies in the input space with SSLAM, including element-wise maximum, average in log-mel-spectrogram, and average in waveform. In our experiments, element-wise max mixing in the log-mel-spectrogram domain performed best (refer to Table 10). Therefore, we employed this approach for partially mixing audio in our work.

Table 10: Comparison of Different Input Audio Mixing Strategies

| Model | mAP |
|---|---|
| **Finetuning** | |
| SSLAM with log-mel-spectrogram element-wise max | 40.9 |
| SSLAM with log-mel-spectrogram average | 40.8 |
| SSLAM with waveform average | 40.4 |

### E.0.2    DIFFERENT AGGREGATION STRATEGIES IN FEATURE SPACE FOR SRL

Table 11: Comparison of Different Feature Aggregation Strategies for SRL

| Model | mAP |
|---|---|
| **Finetuning** | |
| SRL with feature averaging | 40.9 |
| SRL with element-wise max of features | 40.7 |

To generate target representations for SRL, it is necessary to aggregate the features of individual audio clips in feature space. We experimented with two aggregation strategies: averaging and element-wise maximum between the features of two audio clips. Our experiments showed that averaging provided better performance (refer to Table 11). Therefore, we adopted this approach for feature aggregation in our work.

### E.0.3 Visualisation of Mixing via Element-wise Maximum of Log-Mel Spectrograms

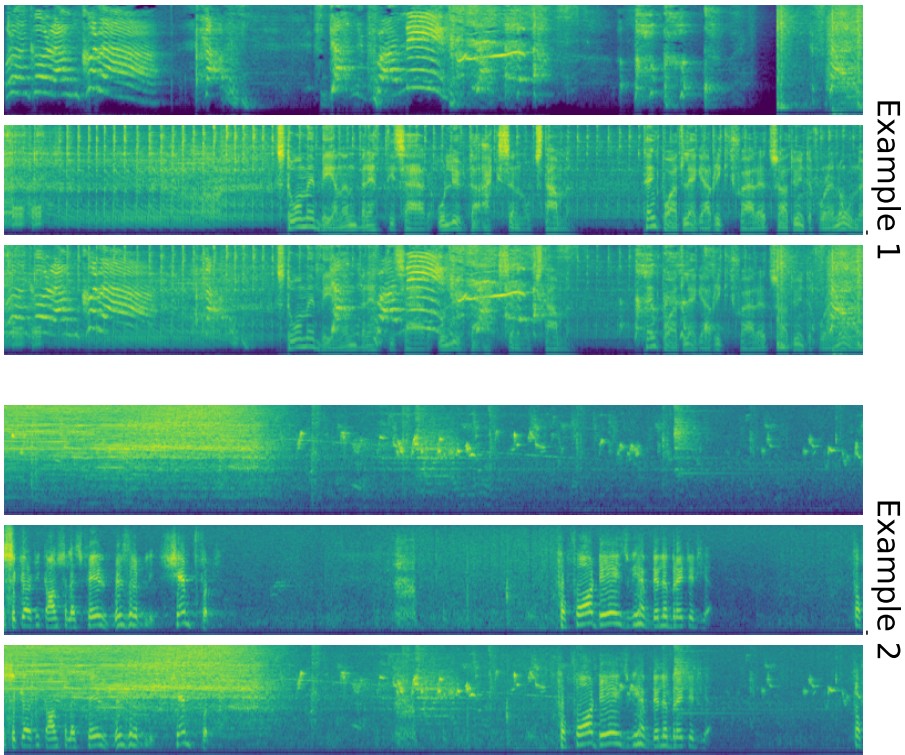

Figure 5: Visualisation of mixing using element-wise maximum of log-mel spectrograms. Each example presents three log-mel-spectrograms: the first two are individual audio samples from AudioSet, and the third is their fully mixed version, created by applying the element-wise maximum of the two spectrograms.

## F  Robustness Analysis Across Multiple Runs

To further validate the robustness of our approach, we conducted three independent runs with different random seeds for the experiments reported in Table 2 and reported the mean and standard deviation in Table 12. These runs specifically aimed to compare our baseline MB-UA with the proposed SSLAM framework, assessing the consistency and reliability of their performance.

As shown in Table 12, the observed trends remain consistent across multiple runs, confirming the robustness of SSLAM and its improvements over MB-UA.

Table 12: Robustness analysis of SSLAM across three independent runs, comparing its performance with the baseline MB-UA. All performances are reported in mAP. For more details about the datasets refer to Appendix B.0.2.

| Model | SPASS | | | | |
|---|---|---|---|---|---|
| | Square | Park | Waterfront | Street | Market |
| **Linear Evaluation** | | | | | |
| MB-UA (Baseline) | $60.15 \pm 0.01$ | $59.72 \pm 0.08$ | $55.23 \pm 0.01$ | $63.50 \pm 0.14$ | $62.69 \pm 0.11$ |
| SSLAM | $64.25 \pm 0.07$ | $64.25 \pm 0.07$ | $59.51 \pm 0.09$ | $67.46 \pm 0.09$ | $68.55 \pm 0.06$ |
| **Fine-tuning** | | | | | |
| MB-UA (Baseline) | $84.40 \pm 0.09$ | $78.38 \pm 0.29$ | $79.81 \pm 0.54$ | $81.47 \pm 0.17$ | $89.91 \pm 0.23$ |
| SSLAM | $85.82 \pm 0.17$ | $80.45 \pm 0.16$ | $82.31 \pm 0.26$ | $82.35 \pm 0.19$ | $90.51 \pm 0.44$ |

