# OpenReview forum: "SSLAM: Enhancing Self-Supervised Models with Audio Mixtures for Polyphonic Soundscapes"
_ICLR.cc/2025/Conference — ICLR 2025 Poster_

### Official Review · Reviewer_dU3n · 2024-10-21

**Soundness:** 4
**Presentation:** 3
**Contribution:** 3
**Rating:** 8
**Confidence:** 5

**Summary:**

In this paper, the authors propose a new self-supervised learning strategy for polyphonic soundscapes, called SSLAM. Unlike traditional masked latent bootstrapping, SSLAM leverages audio mixtures to enhance the self-supervised learning process at both global and local levels. The method demonstrates performance improvements on standard audio self-supervised learning benchmarks, as well as on polyphonic datasets.

**Strengths:**

1. The paper is generally well-written, with clear explanations and logical flow.
2. The paper addresses a fundamental challenge in audio classification—handling the polyphonic nature of real-world audio environments—and proposes a solution based on this insight.
3. SSLAM demonstrates competitive performance on standard audio classification benchmarks.

**Weaknesses:**

1. **Confusing Figure Presentation**: In Figure 1(C), it appears that there are two teacher models, but the description in the text suggests that there is only one teacher model handling two audio inputs separately, which creates some confusion. Additionally, Figure 2 shows layer averaging after the teacher model, whereas in Section 3.1.2, it is stated that only the last layer or all 12 layers are used for global or local loss, which seems inconsistent.
2. **Writing Errors**: There are some citation format issues in Tables 1, 7, and 8, as well as in several paragraphs. Additionally, a numerical error was found in Table 1, where the A-JEPA score for ESC-50 is listed as 06.3 instead of the correct value.

**Questions:**

1. **Audio Mixing**: How does your method compare to commonly used data augmentation techniques, such as Mixup? In Table 4, the performance improvement of mixed audio on AS-20K appears limited, and the baseline results with unmixed audio in stage 1 are already equal or stronger than those of prior works, as shown in Table 1. Could you clarify the specific advantages of your approach in this context?
2. **Layer Averaging**: In section 3.1.2, you mention that averaging over all 12 layers, followed by spatial pooling, could result in excessive information compression. This is why you chose to use only the output of the last layer for the global loss and all 12 layers for the local loss. Could you elaborate on how performance is affected when averaging all 12 layers, as is done in the baseline?
3. **Necessity of Unmixed Objective**: In Tables 2 and 3, the MB-UA-PMA configuration does not show significant improvements over MB-PMA. What would be the impact of using only mixed objectives and SRL? Is it necessary to include the unmixed objectives in this setup?
4. **Polyphonic Audio Datasets**: The datasets you selected for polyphonic scenario evaluation are primarily used for sound event detection, yet you treat them as multi-label classification tasks. Have you considered evaluating performance on sound event detection tasks with time-stamped data? For instance, how would the model perform on SED tasks if an RNN were added after the current model?

---

> ### Author Response · Authors · 2024-11-21
> **Author Response to Reviewer dU3n (1/3)**
>
> **Part 1/3**
>
> ------------
> Dear reviewer dU3n,
>
> Thank you for your positive feedback. Below we address the concerns you raised.
>
> ---------------
>
> **Q1. Confusing Figure Presentation**
>
> Thank you for highlighting this confusion. We will address this question in 2 parts.
>
> >In Figure 1(C), it appears that there are two teacher models, but the description in the text suggests that there is only one teacher model handling two audio inputs separately, which creates some confusion.
>
> We acknowledge the confusion caused by the depiction of two teacher models in Figure 1(C). To clarify, **there is only one teacher model** that processes the two audio inputs separately, as described in the text. While the original figure aimed to visually represent this independent processing, **we have updated the manuscript by revising Figure 1(C) to depict a single teacher model**, ensuring alignment with the text and eliminating any ambiguity.
>
> >Additionally, Figure 2 shows layer averaging after the teacher model, whereas in Section 3.1.2, it is stated that only the last layer or all 12 layers are used for global or local loss, which seems inconsistent.
>
> Thank you for raising this concern. The layer selection for teacher **layer averaging (top-k) differs between mixed and unmixed audio**, as discussed in Table 6 (Top-k Layers) in the Additional Ablations section. For mixed data, top-k=1 is used for the global loss and top-k=12 for the local loss, whereas for unmixed data, top-k=12 is used for both. Additionally, top-k=12 is employed for SRL.
>
> To address this variation, we initially visualized layer averaging as a single block. Based on your suggestion, **we have updated Figure 2 to clearly show that intermediate layers are also used** for layer averaging.
>
> -------------
>
> **Q2. Writing Errors**
>
> >There are some citation format issues ..  correct value.
>
> Thank you for bringing these issues to our attention. We have **corrected the errors and updated the manuscript** accordingly.

---

> ### Author Response · Authors · 2024-11-21
> **Author Response to Reviewer dU3n (2/3)**
>
> **Part 2/3**
>
> _________
> **Q3. Audio Mixing, Mixup, AS-20K performance in stage 1**
>
> Thank you for this question. We will address it in two parts.
>
> >Audio Mixing: How does your method compare to commonly used data augmentation techniques, such as Mixup?
>
> Thank you for bringing up the point about MixUp. It is a widely used data augmentation technique, including in our work, for downstream tasks such as fine-tuning and linear evaluation. As shown in prior works (in Table 1), Mixup improves supervised training by combining audio samples and their corresponding labels. In polyphonic scenarios, it helps expose models to mixed audio, aiding in learning from polyphonic data in a supervised setting. However, **while supervised learning with Mixup can yield useful representations, self-supervised learning is widely recognized for producing more robust and generalized representations.**
>
> In our experiments, both SSLAM and the baseline methods utilized Mixup during downstream tasks, ensuring a fair comparison. SSLAM, however, goes beyond simple data augmentation. It incorporates a self-supervised learning pipeline with novel training objectives, such as Source Retention Loss (SRL), specifically designed to handle polyphonic audio. This enables the model to learn comprehensive polyphonic representations during pre-training, providing it with an edge over baseline approaches that rely solely on supervised learning with Mixup.
>
> The results in **Tables 1, 2, and 3 demonstrate that SSLAM significantly outperforms baseline approaches, even when both are trained with Mixup in downstream tasks**. These results highlight the unique contributions of SSLAM’s self-supervised learning objectives to learning effective polyphonic representations. Hyperparameter details for these experiments are provided in Tables 7 and 8 for reference.
> _________
>
> >In Table 4, the performance improvement of mixed audio on AS-20K appears limited, .. Could you clarify the specific advantages of your approach in this context?
> We will clarify the results based on our understanding of your concern.
>
> As described in the paper, we propose a two-stage training approach:
>
> - Stage 1: The network is trained using unmixed audio data. This stage helps the model learn foundational representations of individual audio events and develop strong feature extraction capabilities without the added complexity of mixed signals. This stage sets the groundwork for effective learning in Stage 2.
>
> - Stage 2: The network is trained using mixed audio data, where we introduce the proposed training objectives to enhance polyphonic learning.
>
> **In Stage 1, we observe that the model trained on mixed audio performs slightly worse than the unmixed baseline.** This performance reduction is expected and is, in fact, a necessary step. It reflects the **model's challenge in learning to process mixed signals at this stage**. However, this performance reduction in Stage 1 sets the stage for a more effective learning process in Stage 2. By first learning on unmixed audio, the model builds the foundational representations needed to handle the complexities of polyphonic data effectively in Stage 2.
>
> Table 4 supports this approach by showing that, in Stage 2, the model trained with mixed audio demonstrates a significant performance improvement. This demonstrates the value of the two-stage training process, where Stage 1 prepares the model for Stage 2, enabling it to learn more effectively from the mixed data.
>
> We hope this explanation clarifies the performance trends in Table 4, particularly in the context of the two-stage approach.
> _____________
>
> **Q4. Layer Averaging Top-k**
>
> >Layer Averaging: In section 3.1.2, you mention that averaging over all 12 layers, ..  elaborate on how performance is affected when averaging all 12 layers, as is done in the baseline?
>
> Thank you for this question. To address this, we refer to the ablation study on top-k layer selection for mixed audio, presented in **Table 6 of the Additional Ablations section**. The results show that using **a higher top-k for the local loss leads to better performance, while a lower top-k value for the global loss performs better**. Specifically, when top-k=12 or top-k=3 is used for the local loss, top-k=1 for the global loss yields better performance compared to using top-k=12 or top-k=3 for the global loss. This ablation helps explain why averaging over all 12 layers, as done in the baseline, may lead to excessive information compression, potentially degrading performance.
>
> To improve clarity on the top-k layer averaging, **we have updated Table 6 to clarify the terminology. Specifically, the terms "CLS loss" and "patch loss" have been standardized throughout the document as "global loss" and "local loss" for consistency.**

---

> > ### Comment · Reviewer_dU3n · 2024-11-25
> >
> > Thank you to the authors for the detailed responses, which address most of my questions. However, I still have a question regarding the AS-20K performance in Stage 1 (Q3). In Table 4, the Stage 1 result with unmixed audio achieves 40.2 mAP on AS-20K, which, as I understand, represents the baseline of the proposed method. Notably, this baseline already exceeds the results of prior works listed in Table 1 (e.g., EAT: 40.2, ASiT: 38.6).
> >
> > This raises a concern about the influence of your baseline setup on the results, as it may lead to an unfair comparison with prior works. It would be helpful to clarify how much of the observed improvement can be attributed specifically to the proposed method versus differences inherent to the baseline setup itself.

---

> ### Author Response · Authors · 2024-11-21
> **Author Response to Reviewer dU3n (3/3)**
>
> **Part 3/3**
> _____
> **Q5. Necessity of Unmixed Objective**
>
> >In Tables 2 and 3, the MB-UA-PMA configuration does not show significant improvements over MB-PMA. ... Is it necessary to include the unmixed objectives in this setup?
>
> Thank you for this observation. Since the sub-questions are closely related, we will address them together.
>
> As discussed in the manuscript, the performance reduction for polyphonic audio in the MB-UA-PMA compared to MB-PMA is expected. In MB-PMA, the full batch consists of mixed data, while in MB-UA-PMA, **half of the batch is mixed audio and the other half is unmixed audio. This reduces the model's exposure to polyphonic data in MB-UA-PMA**, which leads to the observed performance reduction on polyphonic tasks.
>
> Regarding the inclusion of unmixed objectives, we believe their presence is beneficial. The inclusion of unmixed audio helps SSLAM become more robust on monophonic datasets, as demonstrated by the **performance improvements on datasets like ESC-50, KS2, and AS-20K (refer to Appendix D).**  Additionally, unmixed audio allows the network to learn foundational representations of distinct audio events and develop robust feature extraction capabilities, without the added complexity of mixed signals as seen in Stage 1. This ultimately contributes to improved performance on both polyphonic and monophonic tasks.
>
> Furthermore, please note that using two halves of the batch (mixed and unmixed audio) enables a non-redundant and efficient calculation of all five training objectives, particularly SRL, as described in Algorithm 1 in the manuscript.
>
> Although this is already discussed in the manuscript, **we have added a section in the Appendix (Appendix D), along with additional experiments**, to provide further clarification.
>
> _______
>
> **Q6. Polyphonic Audio Datasets**
>
> >The datasets you selected for polyphonic scenario evaluation are primarily used for sound event detection, ..how would the model perform on SED tasks if an RNN were added after the current model?
>
> Yes, you are correct. When incorporating polyphonic audio datasets into the audio SSL evaluation suite, our goal was to assess whether the model could understand polyphonic audio and represent different events within it. We adopted multi-label classification as a means to evaluate whether the learned representations capture multiple audio events, aligning with the standard audio SSL evaluation protocols.
>
> We agree that temporal detection of audio events could provide an additional way to evaluate whether the representations encode multiple audio events. Based on existing evidence, we believe SSLAM is well-suited for this purpose. Audio SSL pre-trained encoders, such as BEATs and ATST, have been widely used and have achieved top rankings [1,2] in SED challenges like DCASE Task 4. Since SSLAM significantly outperforms these methods in our evaluations, we anticipate it would also deliver superior performance on SED tasks.
>
> ___________
>
> We sincerely appreciate your positive feedback and valuable suggestions. We kindly ask you to **review the updated manuscript, which includes the changes discussed above.**
>
> ______
>
> References:
>
> [1]Schmid, F., Primus, P., Morocutti, T., Greif, J., & Widmer, G. (2024). IMPROVING AUDIO SPECTROGRAM TRANSFORMERS FOR SOUND EVENT DETECTION THROUGH MULTI-STAGE TRAINING [Techreport]. DCASE2024 Challenge.
>
> [2]Lyu, H., & He, Q. (2024). Semi-Supervised Sound Event Detection System Based on Complex Convolutional Recurrent Neural Network [Techreport]. DCASE2024 Challenge.

---

> ### Author Response · Authors · 2024-11-25
>
> Dear Reviewer dU3n,
>
> **Baseline training setup**
>
> Please note that we have reproduced the results of EAT (40.2) and ASiT (38.6) from scratch, including both pre-training and fine-tuning in our baseline training setup. That is,  we were able to replicate the reported performance from their respective papers in our baseline setup, without achieving any improvements. This suggests that there is **no unfair inherent advantage in our baseline setup.**
>
> We are very grateful for raising this concern and truly appreciate the importance of fair comparison to progress the field. We are grateful for the insights and valuable feedback which has already improved the manuscript. If there are any further suggestions that could enhance our work, we would be thankful.
>
> Best regards,
>
> The Authors

---

### Official Review · Reviewer_q4mh · 2024-10-28

**Soundness:** 2
**Presentation:** 2
**Contribution:** 2
**Rating:** 6
**Confidence:** 4

**Summary:**

This paper proposes SSLAM, an approach to integrate audio mixing and self-supervised learning, an approach that stems from the fundamental concept that real-world audio data is, generally, polyphonic. The method claims that training on audio mixtures in masked latent bootstrapping framework improves performance for both monophonic and polyphonic soundscapes.

**Strengths:**

1. Clarity: paper is well written and apart from some sections, quite easy to read.
2. Soundness seems sufficient.

**Weaknesses:**

1. Small set of evaluated tasks: not enough diversity in evaluated downstream tasks. Only speech (keyword spotting) and in-domain audio classification tasks are evaluated. Instead of evaluating KS2 and KS1, either one would've sufficed. Dataset choice is also not motivated well enough.
2. Evaluation itself: no mean/std or confidence intervals are reported, which would be even more useful for polyphonic evaluations. Are the downstream results reported from a single test run?
3. The overall objective function in the final SSLAM model is a composite of 5 losses. However, the losses don't seem to be motivated well enough. For instance, the global loss for the mixed version uses only the final layer outputs, which is not the case for your baseline.

**Questions:**

1. Why does SSLAM require both unmixed and mixed versions of the objectives?
2. Why is Table 1 missing the other variants you have evaluated: MB-UA (which I suppose corresponds to the "baseline" you talk about in Section 3.1), MB-PMA, MB-UA-PMA?
3. In table 2, MB-PMA works the best for IDMT DESED in linear evaluation and on SPASS-Market when fine-tuning, but it's not the one highlighted.


POST REBUTTAL
--------------------

Most of my questions have been addressed, so I am raising my recommendation to 6.

---

> ### Author Response · Authors · 2024-11-21
> **Author Response to Reviewer q4mh (1/3)**
>
> **Part 1/3**
> ___
> Dear reviewer q4mh,
>
> Thank you for taking the time to read our work and for your constructive criticisms, which will help us improve it. We appreciate your thoughtful feedback and hope to address the concerns you raised below.
>
> ___
>
> **Q1. Dataset choice is also not motivated well enough**
>
> >Small set of evaluated tasks: not enough diversity in evaluated downstream tasks. ..not motivated well enough.
>
> Thank you for your question. Our dataset choice is primarily motivated by the evaluation suite established by prior works in the field of general audio self-supervised learning (SSL), such as BEATs, AudioMAE. This provides consistency with previous research, allowing us to benchmark our results effectively. Additionally, our approach introduces a novel contribution in the form of enhanced polyphonic data understanding, and we incorporated various polyphonic datasets into the evaluation suite to support this.
>
> Regarding the use of both KS1 and KS2, we acknowledge that the main difference between them is the number of audio files and classes. Prior work has either used one of these datasets or both (refer to Table 1 in the manuscript). By including both, we are able to make more meaningful comparisons with a wider range of existing studies, some of which have used both datasets. We appreciate your understanding of this approach.
>
> ___
> **Q2. Evaluation runs**
>
> >Evaluation itself: no mean/std or confidence intervals are reported, ..test run?
>
> Thank you for the suggestion. Similar to previous audio-SSL works that we compare against, we use a single run and report performance without mean or standard deviation values. We agree with the reviewer that including such statistical measures, particularly for polyphonic evaluations, would offer more insight into the variability of the results.
>
> However, we would like to emphasize that the robustness of our method has been empirically validated across a wide range of polyphonic evaluations. Specifically, we conducted over 120 experiments to assess each incremental component (from MB-UA to SSLAM) across 8 datasets and 7 subsets with varying levels of polyphony, in both linear and fine-tuning setups. The consistent improvements observed across these experiments provide strong evidence supporting the reliability of our approach.
>
> Additionally, in response to your query, **we did perform 3 runs each with different random seeds for a subset of the experiments reported in Table 2 and report mean and standard deviation below**. The results from these runs demonstrate that the performance improvements are consistent. While we did not include these additional runs in the paper to maintain the uniformity of our reporting, we hope that this detailed validation, along with the results from our experiments, offers sufficient confidence in the robustness of our findings.
>
> | Model         | SQARE   | PARK    | WATERFRONT | STREET  | MARKET  |
> |---------------|---------|---------|------------|---------|---------|
> | **Linear Evaluation** |         |         |            |         |         |
> | MB-UA (Baseline)   | 60.15 ± 0.01 | 59.72 ± 0.08 | 55.23 ± 0.01 | 63.50 ± 0.14 | 62.69 ± 0.11 |
> | SSLAM              | 64.25 ± 0.07 | 64.25 ± 0.07 | 59.51 ± 0.09 | 67.46 ± 0.09 | 68.55 ± 0.06 |
> | **Finetuning**     |         |         |            |         |         |
> | MB-UA (Baseline)   | 84.40 ± 0.09 | 78.38 ± 0.29 | 79.81 ± 0.54 | 81.47 ± 0.17 | 89.91 ± 0.23 |
> | SSLAM              | 85.82 ± 0.17 | 80.45 ± 0.16 | 82.31 ± 0.26 | 82.35 ± 0.19 | 90.51 ± 0.44 |

---

> ### Author Response · Authors · 2024-11-21
> **Author Response to Reviewer q4mh (2/3)**
>
> **Part 2/3**
>
> ------
> **Q3. Losses don't seem to be motivated well enough and Top-k layer selection for global loss**
>
> Thank you for this question. We will try to address this question in 2 parts.
>
> >The overall objective function in the final SSLAM model is a composite of 5 losses. However, the losses don't seem to be motivated well enough.
>
> Motivation of the 5 losses is as follows,
>
> **Global and local losses using unmixed audio:** In Stage 1, it helps the network to learn foundational representations of distinct audio events and establish robust feature extraction capabilities without the added complexity of mixed signals.
>
> The reasoning behind retaining un-mixed audio in stage 2 is the following,
>
> - In the MB-PMA model, the training exclusively involves mixed data, without any exposure to unmixed data. We hypothesized that incorporating unmixed data during training could potentially make SSLAM more robust on monophonic datasets. To explore this, we introduced the MB-UA-PMA variant, where each batch(B) contains a mix of mixed audio(B/2) and unmixed audio(B/2). This adjustment demonstrates improved performance on monophonic datasets such as ESC-50, KS2, and AS-20K(partly monophonic)(refer to Table 9 in the manuscript), albeit with a trade-off in performance on polyphonic datasets (refer to Table 2 in the manuscript).
>
> This incorporation was seamless using our Algorithm 1 (in the manuscript), as all parts of the batches could be utilized for different training objectives, such as SRL, without any redundant computation.
>
> Although this reasoning is already discussed in the manuscript, **we have added a section in Appendix (Appendix D) along with additional experiments** to further clarify the rationale behind incorporating unmixed data alongside mixed data during stage 2 of the pre-retraining process.
>
> **Global and local losses with mixed audio:** Exposing the model to mixed audio increases its exposure to diverse polyphonic data, which in turn leads to improved performance on polyphonic tasks, as demonstrated by the performance improvement of MB-PMA over MB-UA in Tables 2 and 3.
>
> **Source retention loss(SRL):**  SRL explicitly enforce represention of mixed audio to be represenative of the source audios. This improves the network's ability in polyphonic audio understanding, as evidenced by the performance of SSLAM over other variants in Tables 2 and 3.
>
>
> >For instance, the global loss for the mixed version uses only the final layer outputs, which is not the case for your baseline.
>
> As shown in Table 6 (Top-k Layers) in Additional Ablations section, we observed that using only the final layer (top-k = 1) for the global loss and top-k = 12 for the local loss yielded the best performance for mixed audio. This differs from the baseline, where top-k = 12 was effective for both losses. It is noteworthy that prior work, such as Data2Vec, suggests that the optimal teacher layer selection (top-k) can vary across modalities, including image, speech, and text, indicating that such selection can also depend on the nature of the input data (e.g., mixed vs unmixed).
>
> To improve clarity on the top-k layer averaging, **we have updated Table 6 to clarify the terminology. Specifically, the terms "CLS loss" and "patch loss" have been standardized throughout the document as "global loss" and "local loss" for consistency.**
>
> _____
>
> **Q4. Need for both unmixed and mixed versions of the objectives**
>
> >Why does SSLAM require both unmixed and mixed versions of the objectives?
>
> In SSLAM, during Stage 2, we use half the batch with mixed data and the other half with unmixed data, allowing us to efficiently calculate all five training objectives using Algorithm 1.
>
> The mixed versions of the objectives expose the model to diverse polyphonic data, which enhances its ability to handle polyphonic tasks. This increased exposure leads to improved performance on polyphonic datasets, as shown by the performance improvements of MB-PMA over MB-UA in Tables 2 and 3.
>
> The inclusion of unmixed data in the training objectives helps SSLAM become more robust on monophonic datasets, as demonstrated by performance improvements on datasets like ESC-50, KS2, and AS-20K **(Appendix D)**. Additionally, it allows the network to learn foundational representations of distinct audio events and develop robust feature extraction capabilities, without the added complexity of mixed signals as seen in Stage 1, ultimately leading to improved performance on both polyphonic and monophonic audio.
>
> Also, for calculating the SRL objective, which further strengthens polyphonic audio understanding, both mixed and unmixed versions of the batch are necessary.

---

> ### Author Response · Authors · 2024-11-21
> **Author Response to Reviewer q4mh (3/3)**
>
> **Part 3/3**
>
> -----
> **Q5. Evaluation of SSLAM variants**
>
> >Why is Table 1 missing the other variants you have evaluated: MB-UA (which I suppose corresponds to the "baseline" you talk about in Section 3.1), MB-PMA, MB-UA-PMA?
>
> Thank you for this question. Our experimental plan was designed to incrementally introduce each novel component, allowing us to evaluate its specific contribution to polyphonic data understanding as we progress toward building the final SSLAM model. To achieve this, we evaluated incremental variants MB-UA, MB-PMA, MB-UA-PMA, and finally SSLAM on polyphonic datasets, **including those referenced in Table 1 (e.g., AudioSet, last column in Table 2)** and datasets with different degrees of polyphony. These results are presented in Tables 2 and 3.
>
> Table 1, on the other hand, focuses on evaluating the robustness of the final model (SSLAM) on the AudioSSL benchmark through a comparative analysis with prior works.
>
> We kindly direct your attention to **Appendix D**, where we compare MB-PMA and MB-UA-PMA on datasets like ESC-50 ,KS2, AS-20K. We hope this addresses your concern regarding the evaluation of these variants. Thank you for considering this addition.
>
> --------
>
> **Q6. Correction to Table 2 Highlighting**
>
> >In Table 2, MB-PMA works the best for IDMT DESED in linear evaluation and on SPASS-Market when fine-tuning, but it's not the one highlighted.
>
> Thank you for pointing that out. We will address this in the revised manuscript.
>
> -------
>
> Thank you again for your valuable feedback. We kindly ask you to review the updated manuscript with the revisions outlined above.

---

> > ### Comment · Reviewer_q4mh · 2024-11-25
> > **Response to author's rebuttal**
> >
> > Thanks for the detailed rebuttal! I appreciate the effort that went into the rebuttal and also appreciate the changes proposed by the authors, as well as the additional experiments for Table 3 (which I recommend you include in the main text).
> >
> > | In SSLAM, during Stage 2, we use half the batch with mixed data and the other half with unmixed data, allowing us to efficiently calculate all five training objectives using Algorithm 1.
> >
> > I recommend you mention this in the main text.
> >
> > I also recommend you add the motivation for the different loss functions in the paper, preferably in the main body but appendices would be fine too.
> >
> > Since a lot of my concerns have been addressed reasonably, I am raising my score to 6.

---

> > > ### Author Response · Authors · 2024-11-26
> > >
> > > Dear Reviewer q4mh,
> > >
> > > Thank you for your constructive feedback. Based on your suggestions, we have made the following revisions to the manuscript.
> > >
> > > >as well as the additional experiments for Table 3 (which I recommend you include in the main text).
> > >
> > > We appreciate your suggestion to include the results of the multiple runs in the main text for Table 2 (We believe the mention of Table 3 was intended to reference Table 2). To ensure uniformity within the manuscript and alignment with the reporting practices of other SSL methods we compare against, we have included the results of these additional runs in the appendix rather than the main text. We believe this addition provides valuable evidence of SSLAM's robustness. These results are now **included under Appendix  F "Robustness Analysis Across Multiple Runs"** in the updated manuscript.
> > >
> > >
> > > >| In SSLAM, during Stage 2, we use half the batch with mixed data and the other half with unmixed data, allowing us to efficiently calculate all five training objectives using Algorithm 1.I recommend you mention this in the main text.
> > >
> > > >I also recommend you add the motivation for the different loss functions in the paper, preferably in the main body but appendices would be fine too.
> > >
> > > We have incorporated these two into the **main text, specifically in Section 3.3, titled "UNIFIED LEARNING FRAMEWORK."**
> > >
> > > Thank you for the thoughtful feedback, which has significantly contributed to improving the manuscript. We welcome any additional suggestions that could further refine our work.
> > >
> > > Best regards,
> > >
> > > The Authors

---

### Official Review · Reviewer_oY15 · 2024-11-01

**Soundness:** 2
**Presentation:** 2
**Contribution:** 2
**Rating:** 6
**Confidence:** 3

**Summary:**

This paper introduces **SSLAM (Self-Supervised Learning from Audio Mixtures)**, a novel self-supervised pre-training strategy for enhancing the performance of audio models on polyphonic soundscapes. Recognizing the limitation of current audio SSL methods, which are often benchmarked on predominantly monophonic datasets, SSLAM incorporates audio mixtures into the pre-training process. The approach utilizes a masked latent bootstrapping framework where a student model is trained on mixtures of audio spectrograms, created via an element-wise maximum operation inspired by the Ideal Binary Mask. Concurrently, a teacher model processes the individual audio sources separately, and its aggregated feature representations serve as targets for the student. A novel source retention loss is introduced to further encourage the model to learn and retain distinct features of each source within the mixture. Experiments on standard audio SSL benchmark datasets (AS-2M, AS-20K, ESC-50, KS1, KS2) demonstrate that SSLAM not only improves performance on polyphonic audio (achieving up to a 9.1% improvement on SPASS and setting a new SOTA mAP of 50.2 on AudioSet-2M) but also maintains or exceeds performance on monophonic datasets. Ablations study the contribution of individual components and the impact of mixing strategies and loss functions.

**Strengths:**

1. The paper tackles the under-explored issue of polyphonic sound processing in self-supervised audio learning. This is crucial because real-world audio scenes rarely consist of isolated sounds, and models trained primarily on monophonic data may struggle to generalize effectively in realistic scenarios.

2. SSLAM introduces a novel training strategy by incorporating audio mixtures and a source retention loss, both well-motivated by principles of auditory scene analysis (specifically, the Ideal Binary Mask concept). The partial mixing strategy further demonstrates a nuanced understanding of the balance between introducing new information and preserving existing audio characteristics.

3. The paper presents convincing empirical results, demonstrating state-of-the-art performance on both polyphonic datasets and standard (largely monophonic) audio SSL benchmarks. The ablation studies and analysis of mixing and aggregation strategies provide further insights into the effectiveness and robustness of the proposed method.

**Weaknesses:**

1. The core contribution of SSLAM, training with mixtures of mixtures, closely resembles the MixIT [A] approach for unsupervised sound separation. The novelty seemingly lies in applying this concept within a self-supervised representation learning framework. However, the paper does not sufficiently justify why this adaptation is novel or contributes significant new insights beyond the well-established principles of mixture invariant training.

2. A major weakness is the absence of a comparison with a straightforward baseline: pre-processing the mixed datasets with an unsupervised sound separation model like MixIT [A] to obtain separated sources, and then applying the self-distillation method. This would directly address the polyphony challenge and provide a more meaningful assessment of the value added by the proposed synthetic mixing strategy.

3. The reliance on a pre-trained model (here, an EMA teacher) trained on potentially millions of labeled audio clips undermines the claim of a fully self-supervised approach. While the student model is trained without labels, the teacher network embodies prior knowledge acquired through supervision. The paper should clearly acknowledge this dependency and avoid potentially misleading terminology. Please revise the manuscript accordingly to reflect that.

4. Figure 4, intended to visualize the mixing process, is of extremely low quality and uses an unconventional orientation (high frequencies displayed at the bottom). This hinders understanding and should be replaced with a lossless image with a correctly oriented frequency axis (low frequencies at the bottom).

5. Despite suggesting broader applicability, the evaluation primarily focuses on sound event detection. While performance gains are demonstrated on polyphonic SED datasets, the paper does not explore other downstream tasks like speech recognition or music analysis where polyphony is also prevalent. This limits the evidence for the generalizability of the learned representations.

[A] Wisdom, S., Tzinis, E., Erdogan, H., Weiss, R., Wilson, K. and Hershey, J., 2020. Unsupervised sound separation using mixture invariant training. Advances in neural information processing systems, 33, pp.3846-3857.

**Questions:**

The paper uses the term "self-supervised," yet the approach relies on a pre-trained teacher model that presumably required supervision. How many labeled audio clips were used to train the teacher model? Can the authors clarify the extent to which their method truly qualifies as self-supervised, given this dependency on a pre-trained, supervised component?  Would "teacher-student training" or a similar term be more accurate?

---

> ### Author Response · Authors · 2024-11-21
> **Author Response to Reviewer oY15 (1/2)**
>
> **Part 1/2**
>
> Dear reviewer oY15,
>
> Thank you for your thoughtful and detailed review of our submission. We appreciate the effort you put into analyzing our work and the insightful comparisons you drew with MixIT. However, we believe there may be some misunderstandings regarding certain aspects of our approach, particularly with respect to EMA teacher self-distillation, the classification of our work as SSL, and the comparison with MixIT. These points may have contributed to some of the critiques and questions raised. Below, we address the concerns outlined under weaknesses and questions.
>
> ______
>
> **Q1. SSLAM vs MixIT, a comparision**
>
> >The core contribution of SSLAM, training with mixtures of mixtures, closely resembles the MixIT [A] approach .. principles of mixture invariant training.
>
> Thank you for your insightful question. While both SSLAM and MixIT (or most of the source separation approaches, for that matter) involve mixing audio inputs, their underlying methodologies and objectives are fundamentally different. SSLAM and MixIT differ in both what they aim to achieve and how they accomplish it.
>
> As is evident, the two have distinct objectives (representation learning vs. audio source separation). Beyond the shared idea of mixing audio inputs, how SSLAM learns representations is fundamentally different from how MixIT performs audio source separation. Specifically, SSLAM doesn't use mixture invariant training loss.
>
> We hope the following explanation clarifies the distinction:
>
> **MixIT and Source Separation**: In audio source separation, creating mixtures from isolated (monophonic) sources and using them as targets is a common approach. However, this requires isolated inputs, which are often unavailable in real-world scenarios. MixIT overcomes this limitation by applying mixture invariant training, enabling the use of non-isolated audio inputs. By mixing non-isolated sources, MixIT generates "mixtures of mixtures" and trains a convolutional network to separate them into individual waveforms, leveraging the non-isolated sources to compute the loss via the mixture invariant training strategy.
>
>  **SSLAM and Audio SSL vs MixIT** : Making a comparison between SSLAM and MixIT is difficult as they are different in what they do and how they do it like we discussed above. However, there are key differences in terms of architecture (CNN vs Transformer-based Teacher-Student model), targets (input waveforms vs EMA teacher outputs), training objectives (mixture invariant loss vs SSLAM objectives like SRL), and the output space (waveform space vs latent space). Fundamentally, as you mentioned, MixIT focuses on separating individual sound sources, whereas SSLAM is designed to learn combined representation of the mixed audio.
>
> We hope your next question regarding the baseline using MixIT will provide further clarity on this.
>
> ______
>
> **Q2. Baseline using MixIT**
> > A major weakness is the absence of a comparison with a straightforward baseline: pre-processing the mixed datasets with an unsupervised sound separation model like MixIT [A] to obtain separated sources .. by the proposed synthetic mixing strategy.
>
> Thank you for this valuable suggestion. We understand your concern regarding the comparison with a baseline involving pre-processing using an unsupervised sound separation model like MixIT. However, applying MixIT directly as a baseline is not entirely straightforward, as MixIT produces separated audio waveforms, while SSLAM requires the encoder to output audio representations rather than separated signals.
>
> That said, we agree with your point that concepts from mixture-invariant training (such as separating sources from mixtures) can inspire self-supervised representation learning. Specifically, **the idea of learning representations by separating concepts or representations from mixed audio representation** could be explored further.
>
> In fact, we conducted an experiment based on this idea during the development of SSLAM. **This approach, inspired by MixIT, achieved a performance of 39.9 mAP on AudioSet 20K, which is lower than SSLAM's 40.9 mAP**. We did not include this experiment in the original manuscript as it did not fit seamlessly into the narrative and could potentially confuse readers unfamiliar with source separation techniques.
>
> However, based on your suggestion, we have now added **this experiment to the Appendix (refer to Appendix C)** to provide a more complete context for this comparison. Thank you for helping us improve the clarity and scope of the work.

---

> ### Author Response · Authors · 2024-11-21
> **Author Response to Reviewer oY15 (2/2)**
>
> **Part 2/2**
>
> __________
> **Q3. Teacher model is pre-trained?**
>
> >The reliance on a pre-trained model (here, an EMA teacher) trained on potentially millions of labeled audio clips undermines the claim of a fully self-supervised approach. .. reflect that.
>
> Thank you for this question. We would like to clarify that **we do not use any pre-trained teacher model or labeled data in our pre-training**. Instead, the teacher model is an exponential moving average (EMA) of the student model, following EAT, Data2Vec and prominent vision SSL works such as DINO[1], BYOL[2] etc.The teacher's parameters, $\theta_t$, are updated according to the EMA rule:
>
> $\theta_t \leftarrow \tau \theta_t + (1 - \tau) \theta_s$
>
> where $\theta_s$ denotes the parameters of the student model. We used the term self-distillation rather than distillation in Figure 1 to emphasize this. Additionally, **we have added a stop gradient indicator on the teacher model in Figure 2.** We hope this serves as an additional feature to make our work more clearly identifiable as an EMA teacher-based self-distillation approach.
> ___
> **Q4. Improving the Mixing visualization**
>
> >Figure 4, intended to visualize the mixing process, is of extremely low quality  ...(low frequencies at the bottom).
>
> Thank you for pointing that out. **We have updated the figure in the manuscript (it is now Figure 5).**
> ___
> **Q5. SSLAM's generalization and evaluation datasets**
>
> >Despite suggesting broader applicability, the evaluation primarily focuses on sound event detection. .. the learned representations.
>
> We thank you for the constructive suggestion. While we agree that evaluating SSLAM on additional downstream tasks such as speech recognition or music analysis would provide valuable insights, our current work primarily focuses on sound event detection (SED) because our approach aligns with the broader general audio self-supervised learning (SSL) framework, similar to previous works like BEATs, AudioMAE  etc. which are evaluated on evaluation set consists of general audio, environmental audio, and speech. This benchmark suite has been widely adopted in the community and the these networks are widely adopted as a general audio encoder in other works, further demonstrating the general applicability of the learned representations.
>
> As the reviewer may be aware, different SSL approaches are often tailored to specific domains, such as MERT[3], which is pre-trained on music datasets, or wav2vec[4] and HuBERT[5], which use dedicated speech datasets. Direct comparisons between our general-purpose SSL approach and these domain-specific models would not be entirely fair, as they are optimized for distinct tasks. We believe this approach has broader applicability in audio representation learning, but we recognize that expanding our evaluation to additional tasks such as speech recognition or music analysis would further demonstrate the versatility of SSLAM.
> ___
> **Q6. Usage of term "self-supervised"**
>
> > The paper uses the term "self-supervised," yet the approach relies on a pre-trained teacher ..similar term be more accurate?
>
> As mentioned in the response to **Q3**, we do not use a pre-trained teacher or labeled data in our pre-training; instead, we use the exponential moving average (EMA) of the student model as the teacher. Therefore, **this is a “self-supervised” approach.**
>
> ___
> We thank you again for the detailed review and constructive suggestions. We request you to **consider the updated manuscript with the updates mentioned above.**
>
> ___
> References:
>
> [1]Caron, M., Touvron, H., Misra, I., Jégou, H., Mairal, J., Bojanowski, P., & Joulin, A. (2021). Emerging properties in self-supervised vision transformers. Proceedings of the IEEE/CVF International Conference on Computer Vision, 9650–9660.
>
> [2]Grill, J.-B., Strub, F., Altché, F., Tallec, C., Richemond, P., Buchatskaya, E., Doersch, C., Avila Pires, B., Guo, Z., Gheshlaghi Azar, M., & others. (2020). Bootstrap your own latent-a new approach to self-supervised learning. Advances in Neural Information Processing Systems, 33, 21271–21284.
>
> [3]Li, Y., Yuan, R., Zhang, G., Ma, Y., Chen, X., Yin, H., Lin, C., Ragni, A., Benetos, E., Gyenge, N., & others. (2023). MERT: Acoustic music understanding model with large-scale self-supervised training. ArXiv Preprint ArXiv:2306.00107.
>
> [4]Baevski, A., Zhou, Y., Mohamed, A., & Auli, M. (2020). wav2vec 2.0: A framework for self-supervised learning of speech representations. Advances in Neural Information Processing Systems, 33, 12449–12460.
>
> [5]Hsu, W.-N., Bolte, B., Tsai, Y.-H. H., Lakhotia, K., Salakhutdinov, R., & Mohamed, A. (2021). Hubert: Self-supervised speech representation learning by masked prediction of hidden units. IEEE/ACM Transactions on Audio, Speech, and Language Processing, 29, 3451–3460.

---

> ### Author Response · Authors · 2024-11-25
>
> Dear reviewer oY15,
>
> Thank you for your valuable feedback and for recognizing our efforts in addressing your concerns. We also truly appreciate your decision to increase your score.
>
> >The teacher model sees as input monophonic (single source data) which is why the approach cannto be deemed purely self-supervised (authors claim that in page 2 lines 100-102).
>
> Regarding your observation about the teacher model's input being monophonic, we have made adjustments to clarify this aspect in our manuscript. Specifically, we elaborated on **the self-supervised nature of our approach** by specifying that the audio inputs are selected **randomly from AudioSet which contains both monophonic and polyphonic audios.** This means we do not know in advance whether the selected signals are monophonic or polyphonic, and consequently, the mixed audio could either be a mixture or a mixture of mixtures. The teacher model, on the other hand, can process any given two audio signals, whether they are monophonic or polyphonic, independently. The key idea is that the teacher always processes these two audio inputs separately to ensure independent feature extraction for each.
>
> This clarification has been included in the revised manuscript.
>
> Best regards,
>
> The Authors

---

### Official Review · Reviewer_rVH4 · 2024-11-04

**Soundness:** 4
**Presentation:** 4
**Contribution:** 3
**Rating:** 8
**Confidence:** 3

**Summary:**

This paper presents SSLAM, an SSL framework for audio that incorporates audio mixtures to better adapt to polyphonic environments. SSLAM leverages a two-stage training process: the first stage uses single-source audio, while the second incorporates partially mixed audio to simulate polyphonic scenarios. The paper introduces Source Retention Loss to preserve the distinct characteristics of each audio source in a mixture, making the model more robust in complex, multi-source audio settings. The authors evaluate SSLAM on a mix of monophonic and polyphonic datasets, reporting strong improvements over prior methods.

**Strengths:**

The paper addresses a relevant challenge in audio SSL, especially as polyphonic environments are common in real-world audio. The two-stage training approach combined with the Source Retention Loss appears effective, allowing the model to learn from mixed audio in a way that preserves source integrity. The evaluations include a range of polyphonic datasets, and the method generally performs well across them. The component-wise analysis of the model’s objectives is also informative, showing the impact of each training objective.

**Weaknesses:**

The paper does not provide an analysis of SSLAM’s generalization to unseen audio domains, such as speech or music distribution. It would be great to also observe if this technique is applicable to other domains’ downstream applications such as multi-speaker recognition or instrument recognition, where audio types could differ significantly from pre-training data.

**Questions:**

Since AudioSet is not fully monophonic, what is the intuition behind the mixing technique’s consistent performance improvement in polyphonic tasks? For example, in URBAN-SED, this improvement is not always observed, so further insights into why mixing boosts performance in some cases but not others would be helpful.

---

> ### Author Response · Authors · 2024-11-21
> **Author Response to Reviewer rVH4**
>
> Dear reviewer rVH4,
> We sincerely thank you for the thoughtful evaluation and high score given to our work. Below, we address questions and concerns individually.
> ___
>
> **Q1. SSLAM's generalization to unseen audio domains**
>
> >SSLAM’s generalization to unseen audio domains, such as speech or music distribution. It would be great to also observe if this technique is applicable to other domains’ downstream applications such as multi-speaker recognition or instrument recognition
>
> We thank you for the suggestion and acknowledge your interest in evaluating SSLAM on other audio domains such as music and speech. However, to ensure a comprehensive evaluation, we employed a widely adopted audio SSL benchmark suite, including general audio (a mix of music, speech, traffic, and others), environmental sounds, and speech. This evaluation aligns with prior works, such as BEATs, AudioMAE and others(refer to Table 1 in the manuscript), which have followed similar protocols. Their adoption as audio encoders in various machine-learning systems highlights their potential for generalization. Additionally, we have extended this evaluation with polyphonic datasets to ensure a comprehensive and robust assessment of SSLAM's capabilities.
>
> We appreciate your valuable suggestion to explore unseen and other audio domains benchmarks, such as multi-speaker recognition or instrument identification, as potential applications for SSLAM. However, we note that our work is focused on general audio representation learning rather than domain-specific adaptation. Comparisons with domain-specialized SSL models (e.g., wav2vec [1] and HuBERT[2]  for speech or MERT [3] for music), which are pre-trained and optimized for highly specific datasets, would not provide a fair evaluation of our approach. Such models benefit from tailored datasets and objectives, which are outside the scope of our current contribution.
> ___
> **Q2. Mixing Intuition, Performance on URBAN-SED**
>
> We thank the reviewer for their thoughtful question,  which allows us to elaborate further on the intuition and observed results. Below, we address the question in two parts:
>
> >Since AudioSet is not fully monophonic, what is the intuition behind the mixing technique’s consistent performance improvement in polyphonic tasks?
>
> We apply a random mixing strategy, combining two audio files from AudioSet to create either a straightforward mix of monophonic files or a more complex "mixture of mixtures" when the files are not purely monophonic. This randomness broadens the range of polyphonic scenarios, enriching the diversity of training data beyond what AudioSet alone provides. By exposing the model to these varied audio combinations, it learns more robust and generalizable polyphonic representations, as reflected in the improved performance of MB-PMA over MB-UA on polyphonic tasks (Tables 2 and 3). To better clarify this intuition, we have made a minor addition to the performance discussion Section 5, ("Performance Discussion: Polyphonic Audio Dataset,) of the manuscript. Additionally, the SRL reinforces this improvement by guiding the model to disentangle and retain representations of individual audio components, further enhancing SSLAM’s performance (Tables 1 and 2).
>
> ___
>
> >in URBAN-SED, this improvement is not always observed
>
> Among the polyphonic datasets we used, SPASS (2023) is the most recent and was generated with particular attention to the placement and distribution of sound events within environmental scenes, resulting in higher quality as a polyphonic audio dataset. IDMT (2021) is relatively newer, while URBAN-SED (2017) is the oldest dataset in our evaluation. Additionally, the software and methodologies used for generating SPASS differ from those used for the other datasets. We hypothesize that these differences in quality as polyphonic audio datasets, especially in the older URBAN-SED, contribute to the variations in performance improvements observed.
> ___
>
> We sincerely thank you once again for your thoughtful questions and positive feedback. We would like to inform you that we have made a minor update to the manuscript, as outlined above, to provide additional clarity and address your question.
> ___
> References:
>
> [1]Baevski, A., Zhou, Y., Mohamed, A., & Auli, M. (2020). wav2vec 2.0: A framework for self-supervised learning of speech representations. Advances in Neural Information Processing Systems, 33, 12449–12460.
>
> [2]Hsu, W.-N., Bolte, B., Tsai, Y.-H. H., Lakhotia, K., Salakhutdinov, R., & Mohamed, A. (2021). Hubert: Self-supervised speech representation learning by masked prediction of hidden units. IEEE/ACM Transactions on Audio, Speech, and Language Processing, 29, 3451–3460.
>
> [3]Li, Y., Yuan, R., Zhang, G., Ma, Y., Chen, X., Yin, H., Lin, C., Ragni, A., Benetos, E., Gyenge, N., & others. (2023). MERT: Acoustic music understanding model with large-scale self-supervised training. ArXiv Preprint ArXiv:2306.00107.

---

> > ### Comment · Reviewer_rVH4 · 2024-11-27
> >
> > Thank you for your response. The rebuttal addressed my concerns from the initial review, and I maintain my recommendation for accepting the paper.

---

### Author Response · Authors · 2024-11-21
**Author Response and Manuscript Revisions**

Dear Reviewers,

Thank you for your detailed analysis and constructive suggestions on our paper, SSLAM: Enhancing Self-Supervised Models with Audio Mixtures for Polyphonic Soundscapes.

In response to your feedback and to improve the clarity of specific aspects of our work, we have updated the manuscript accordingly. The revised PDF has been uploaded, and we kindly ask you to review it. Below is a summary of the changes made for your convenience:

- Reviewer **rVH4**: To provide additional clarity on the intuition behind the consistent performance improvements of the mixing technique, we have made a minor addition to Section 5, "Performance Discussion: Polyphonic Audio Dataset."

- Reviewer **oY15**: In response to your suggestion, we have made the following revisions: (1) added a new section in the Appendix (Appendix C) that explores *Representation Learning via Concept Separation with Mixture Invariant Loss (MixIT),* (2) updated Figure 2 by adding the "stop gradient" notation to make it identifiable with previous self-distillation SSL works, and (3) improved the quality of the mixing visualization in Figure 5.

- Reviewer **q4mh**: Based on your feedback, we have made the following updates: (1) added a new section in the Appendix (Appendix D) with additional experiments and clarifications regarding the necessity of the unmixed training objective during Stage 2, (2) updated Table 6 to improve clarity on top-k layer averaging and standardized the terminology, specifically changing "CLS loss" and "patch loss" to "global loss" and "local loss" throughout the document, and (3) addressed the highlighting issue in Table 2.

- Reviewer **dU3n**: Based on your feedback, we have made the following updates: (1) revised Figures 1 and 2, (2) corrected the writing errors you pointed out, (3) updated Table 6 to improve clarity on top-k layer averaging and clarified the terminology, specifically standardizing "CLS loss" and "patch loss" as "global loss" and "local loss" throughout the document , and (4) added a new section in the Appendix (Appendix D) with additional experiments and clarifications regarding the necessity of the unmixed training objective during Stage 2.


We believe these updates improve the clarity and completeness of our work, and we look forward to your further feedback. Below, we address each reviewer’s concerns in detail.

---

### Author Response · Authors · 2024-11-24
**Follow-Up on Author Response**

Dear Reviewers,

Thank you for your valuable feedback and for taking the time to review our submission. We have carefully considered and addressed all your comments to the best of our ability and wanted to follow up to confirm if you’ve had a chance to review our response submitted a few days ago. If there’s anything we can clarify or expand upon to further enhance the quality of our work, please let us know.

Thank you,

The Authors

---

### Meta-Review · Area_Chair_ubZc · 2024-12-22

**Metareview:**

**Paper Summary:**

This paper describes a self-supervised approach to modeling multi-source audio signals SSLAM. This is achieved by training on mixtures of audio signals, and using a source retention loss that pushes the model to learn distinct features of each source within a mixture. Experiments show that SSLAM maintains or improves performance on single-source audio classification (e.g., AudioSet) while significantly increasing performance on multi-source audio classification datasets (SPASS, IDMT, URBAN).

**Strengths:**

Reviewers uniformly recognized the importance of the multi-source audio setting. They found the experimental results convincing, which demonstrate superior performance of SSLAM for both single- and multi-source audio datasets.

**Weaknesses:**

Reviewer oY15 raises a concern about methodological similarity to MixIT. This is partially addressed in the author response with a direct comparison to a MixIT-inspired baseline (which SSLAM does outperform).

The paper focuses on sound event detection; Reviewer oY15 also raises concerns about the generality of SSLAM, with resect to other types of audio data: e.g., speech recognition or music analysis. The authors may have invited this criticism by using "polyphony"--a word typically reserved for describing music--to describe their multi-source learning task. Nevertheless, I tend to agree with the authors that a more expansive analysis of these other audio learning tasks is out of scope.

**Additional Comments On Reviewer Discussion:**

Most of the issues raised in the reviews were addressed during the author response periods. There was a lively discussion, leading several reviewers to increase their opinion of this work.

---

### Decision · Program_Chairs · 2025-01-22

Accept (Poster)